# Self-carbon-thermal-reduction strategy for boosting the Fenton-like activity of single Fe-N$_4$ sites by carbon-defect engineering

Shengjie Wei [1,2,7], Yibing Sun [3,7], Yun-Ze Qiu [2,7], Ang Li [4,7], Ching-Yu Chiang [5] ✉, Hai Xiao [2] ✉, Jieshu Qian [3,6] ✉ & Yadong Li [2] ✉

Carbon-defect engineering in metal single-atom catalysts by simple and robust strategy, boosting their catalytic activity, and revealing the carbon defect-catalytic activity relationship are meaningful but challenging. Herein, we report a facile self-carbon-thermal-reduction strategy for carbon-defect engineering of single Fe-N$_4$ sites in ZnO-Carbon nano-reactor, as efficient catalyst in Fenton-like reaction for degradation of phenol. The carbon vacancies are easily constructed adjacent to single Fe-N$_4$ sites during synthesis, facilitating the formation of C-O bonding and lowering the energy barrier of rate-determining-step during degradation of phenol. Consequently, the catalyst Fe-NCv-900 with carbon vacancies exhibits a much improved activity than the Fe-NC-900 without abundant carbon vacancies, with 13.5 times improvement in the first-order rate constant of phenol degradation. The Fe-NCv-900 shows high activity (97% removal ratio of phenol in only 5 min), good recyclability and the wide-ranging pH universality (pH range 3-9). This work not only provides a rational strategy for improving the Fenton-like activity of metal single-atom catalysts, but also deepens the fundamental understanding on how periphery carbon environment affects the property and performance of metal-N$_4$ sites.

Metal-isolated single-atom site (ISAS) catalysts with utmost atomic utilization efficiency exhibit good catalytic performance in heterogeneous catalysis[1–7]. Among them, metal ISAS anchored on carbon-based substrates deriving from Metal-Organic-Frameworks (MOFs) materials have been widely utilized in electrocatalysis[8–13], photocatalysis[14–17], and thermocatalysis[18–21], owing to the excellent conductivity, low-cost, and easy synthesis of carbon-based materials. Optimizing the coordination environment of metal ISAS is the key issue to improve their catalytic performance, such as regulation of coordination numbers[22–25], introduction of hetero-atoms as coordination atoms[26–28], and synergistic effect of different catalytic sites[29].

Particularly, the development of atomic-defect engineering of metal ISAS is a rational strategy to boost catalytic performances, such as nitrogen-vacancy[30–36], sulfur-vacancy[37], and oxygen-vacancy engineering[38–40]. Constructing the nitrogen-vacancy on the first coordination shell of metal ISAS on N-doped carbon substrates can optimize the coordination geometry of metal catalytic sites and boost their activity during catalysis[30–32]. Besides, introducing carbon-vacancy on the second coordination shell of metal-N$_4$@N-doped carbon catalysts can also improve the catalytic performance, such as oxygen reduction reaction[41–44] and CO$_2$ reduction reaction[45]. Currently, the carbon-defect engineering of metal ISAS catalysts for boosting catalytic

[1]School of Materials Science and Engineering, Nankai University, Tianjin 300350, China. [2]Department of Chemistry, Tsinghua University, Beijing 100084, China. [3]Jiangsu Key Laboratory of Chemical Pollution Control and Resources Reuse, School of Environmental and Biological Engineering, Nanjing University of Science and Technology, Nanjing 210094, P. R. China. [4]Faculty of Materials and Manufacturing, Beijing Key Lab of Microstructure and Properties of Advanced Materials, Beijing University of Technology, Beijing 100124, P. R. China. [5]National Synchrotron Radiation Research Center, Hsinchu 30076, Taiwan. [6]School of Environmental Engineering, Wuxi University, Jiangsu 214105, P. R. China. [7]These authors contributed equally: Shengjie Wei, Yibing Sun, Yun-Ze Qiu, Ang Li. ✉e-mail: chiang.cy@nsrrc.org.tw; haixiao@tsinghua.edu.cn; qianjieshu@njust.edu.cn; ydli@mail.tsinghua.edu.cn

activity mainly focuses on electro-catalysis[41–45]. Therefore, it is significant to rationally design carbon-defect engineering of metal ISAS catalysts, to expand the new field of catalysis, to boost the catalytic activity and to reveal the relationship between carbon-defect and catalytic activity. However, skillfully designing carbon-defect engineering by simple and robust strategy and revealing the structure-activity relationship during catalysis are challenging.

In recent years, Fenton reaction and its derived Fenton-like reactions have received tremendous interest in various fields, including biomedicine[46], gene expression[47], biosensing[48], material and chemical synthesis[49,50], and environmental remediation[51]. Particularly, the original version of Fenton reaction using homogeneous $Fe^{2+}$ and $H_2O_2$ has achieved global application in many water treatment facilities because the generated hydroxyl radicals (•OH) with strong oxidizing ability could effectively remove toxic organic pollutants[52]. However, this reaction faces several challenges including non-regenerable catalyst, narrow pH range, formation of Fe sludge, and unsafe manipulation of $H_2O_2$[53]. Consequently, a lot of efforts have been made to develop efficient heterogeneous Fenton catalysts and use alternative oxidants especially peroxymonosulfate (PMS) to address the above issues[54,55]. Various metal ISAS catalysts have exhibited great potential for PMS activation and pollutant removal[56–61], for which optimizing the chemical environment of catalytic sites and revealing the structure-activity relationship are crucial for boosting the catalytic activity. Very recently, several atomic engineering strategies including oxygen vacancy[62], oxygen doping[63], and coordination modulation[64] have been reported to further improve the catalytic activity of the metal ISAS catalysts. However, the effect of carbon-defect engineering of metal ISAS catalysts on Fenton-like reaction remains unexplored, which is also appealing for improving their activity. Therefore, developing simple and low-cost synthetic methodology, rationally designing carbon-defect engineering, boosting Fenton-like activity of metal ISAS, and revealing the structure-activity relationship are meaningful for improved degradation of pollutants but with great challenges.

Herein, we design a carbon-defect engineering of single Fe-$N_4$ sites in ZnO-Carbon nano-reactor via self-carbon-thermal-reduction strategy, as efficient Fenton-like catalyst for degradation of phenol. By a combination of experimental results and density functional theory (DFT) calculations, we reveal that the carbon vacancies are easily constructed adjacent to single Fe-$N_4$ sites, facilitating the formation of C-O bonding and lowering the energy barrier of rate-determining-step during catalysis. Consequently, compared with Fe-NC-900 without abundant carbon vacancies, Fe-NCv-900 with abundant carbon vacancies exhibits a much improved activity, with 13.5 times improvement in the first-order rate constant for degradation of phenol. Besides, Fe-NCv-900 exhibits high activity (97% removal ratio of phenol in only 5 min), good recyclability and wide pH suitability (3-9). This work represents a rational attempt to boost the Fenton-like activity of metal ISAS by carbon-defect engineering and to elucidate the relationship between the carbon-defect and catalytic activity in Fenton-like reaction, which are crucial for improving the degradation efficiency of organic pollutants in water remediation. The simple, cheap, and robust self-carbon-thermal-reduction strategy will hopefully inspire further studies on developing other rational synthetic methodology in carbon-defect engineering. The combination of multiple characterization techniques, such as in situ EXAFS, in situ ETEM, and AC-STEM measurements, provides a strong support to directly observe the structural evolution of catalysts and to reveal the structure-activity relationship during catalysis.

## Results

### Self-carbon-thermal-reduction strategy and characterization

The Fe-NCv catalysts were synthesized by self-carbon-thermal-reduction strategy. The synthetic procedure was illustrated in Fig. 1a and Supplementary Fig. 1. ZIF-8 powder and iron acetylacetonate

(Fe(acac)$_3$) on filter papers, as the precursors of N-doped carbon substrate and Fe ISAS, respectively, were under co-pyrolysis in the argon atmosphere. The characterization of ZIF-8 was exhibited in Supplementary Figs. 2-4. During co-pyrolysis around 340 °C, the $H_2O$ vapor by decomposition of hydroxyl groups from filter papers in situ released, confirmed by thermogravimetric analysis coupled with fourier-transform infrared spectroscopy and mass spectrometry (TG-FTIR-MS) in Supplementary Fig. 5. The in situ released $H_2O$ vapor reacted with the $Zn^{2+}$ ions from ZIF-8, with in situ formation of ZnO NPs as nano-oxidant and collapse of ZIF-8, confirmed by high angle annular dark field scanning transmission electron microscopy (HAADF-STEM) images in Fig. 1 (Fe-NCv-400) and Supplementary Fig. 6. Except for agglomeration of $Zn^{2+}$ ions and formation of ZnO NPs, the volatile Fe species from Fe(acac)$_3$ evaporated and the coordination-unsaturated nitrogen atoms from ZIF-8 efficiently captured the volatile Fe species, with the exchange between $Zn^{2+}$ ions and Fe species. During co-pyrolysis from 400 °C to 600 °C, the ZnO NPs in situ formed and grew larger, with the average diameters of ZnO NPs increasing from 42.0 nm, 44.0 nm, and 53.5 nm at 400 °C, 500 °C, and 600 °C, respectively, confirmed by HAADF-STEM images in Fig. 1b–d (Fe-NCv-400 to Fe-NCv-600) and Supplementary Figs. 6–9. The ZIF-8 collapsed and was carbonized gradually from 400 °C to 600 °C. From 700 °C to 900 °C, the self-carbon-thermal-reduction occurred spontaneously in ZnO-Carbon nano-reactor, which was composed of ZnO NPs as nano-oxidant and carbon-substrate as nano-reductant, with in situ disappearance of ZnO NPs and formation of carbon-defects, measured by HAADF-STEM images in Fig. 1e–g (Fe-NCv-700 to Fe-NCv-900) and Supplementary Figs. 10–12. The X-ray diffraction (XRD) measurements in Supplementary Fig. 13 revealed the carbonization of ZIF-8 above 600 °C, with the vanishment of diffraction peaks of ZIF-8. The XRD diffraction peaks of ZnO existed in Fe-NCv-500 and Fe-NCv-600 while those in situ disappeared in Fe-NCv-700, Fe-NCv-800, and Fe-NCv-900, confirming the in situ formation and in situ disappearance of ZnO NPs. The contents of Fe and Zn elements from Fe-NCv catalysts were listed in Supplementary Table 1 by inductively coupled plasma optical emission spectrometry (ICP-OES) measurements. The Zn contents decreased sharply above 600 °C during the in situ disappearance of ZnO NPs. The Fe and Zn contents of Fe-NCv-900 were 0.75 wt% Fe and 1.67 wt% Zn, respectively.

We directly observed the existence form of metal from Fe-NCv catalysts by AC-STEM measurements. In Fig. 2a, the ZnO NPs existed in Fe-NCv-600, with the lattice spacing of 0.281 nm, corresponding to ZnO (100) surface. The metal ISAS also dispersed on the substrate of Fe-NCv-600 in Fig. 2b. After in situ disappearance of ZnO NPs by self-carbon-thermal-reduction, the metal ISAS solely existed in substrates of Fe-NCv-700, Fe-NCv-800, and Fe-NCv-900 (Figs. 2c-2e). In order to track the structural evolution of ZnO NPs during self-carbon-thermal-reduction, in situ ETEM measurements were performed by heating Fe-NCv-600 under Ar atmosphere. At 500 °C in Fig. 2f, two ZnO NPs from Fe-NCv-600 were marked by white circle and white rectangle, respectively. After heating at 700 °C in Fig. 2g, the Zn-based NPs' contrast became darker. We measured the lattice spacing of Zn-based NPs with darker contrast by AC-STEM image in Fig. 2h, confirming the transformation from ZnO NPs into Zn NPs. The lattice spacing of 0.232 nm and 0.250 nm were ascribed to Zn (100) and (002) surfaces, respectively. Therefore, the self-carbon-thermal-reduction between ZnO and carbon substrate underwent the structural evolution from ZnO NPs into Zn NPs, evaporation of Zn NPs and formation of carbon-defects.

The distributions of elemental species from Fe-NCv catalysts were analyzed by X-ray photoelectron spectroscopy (XPS) measurements in Supplementary Figs. 14–19. In the XPS spectra of O 1$s$, the O-Zn peak was the main component in Fe-NCv-400 and Fe-NCv-500 and gradually decreased from Fe-NCv-600 to Fe-NCv-900 during in situ disappearance of ZnO NPs. In the XPS spectra of N 1$s$ from Fe-NCv-400 to Fe-NCv-600, the proportion of N-metal peak decreased obviously

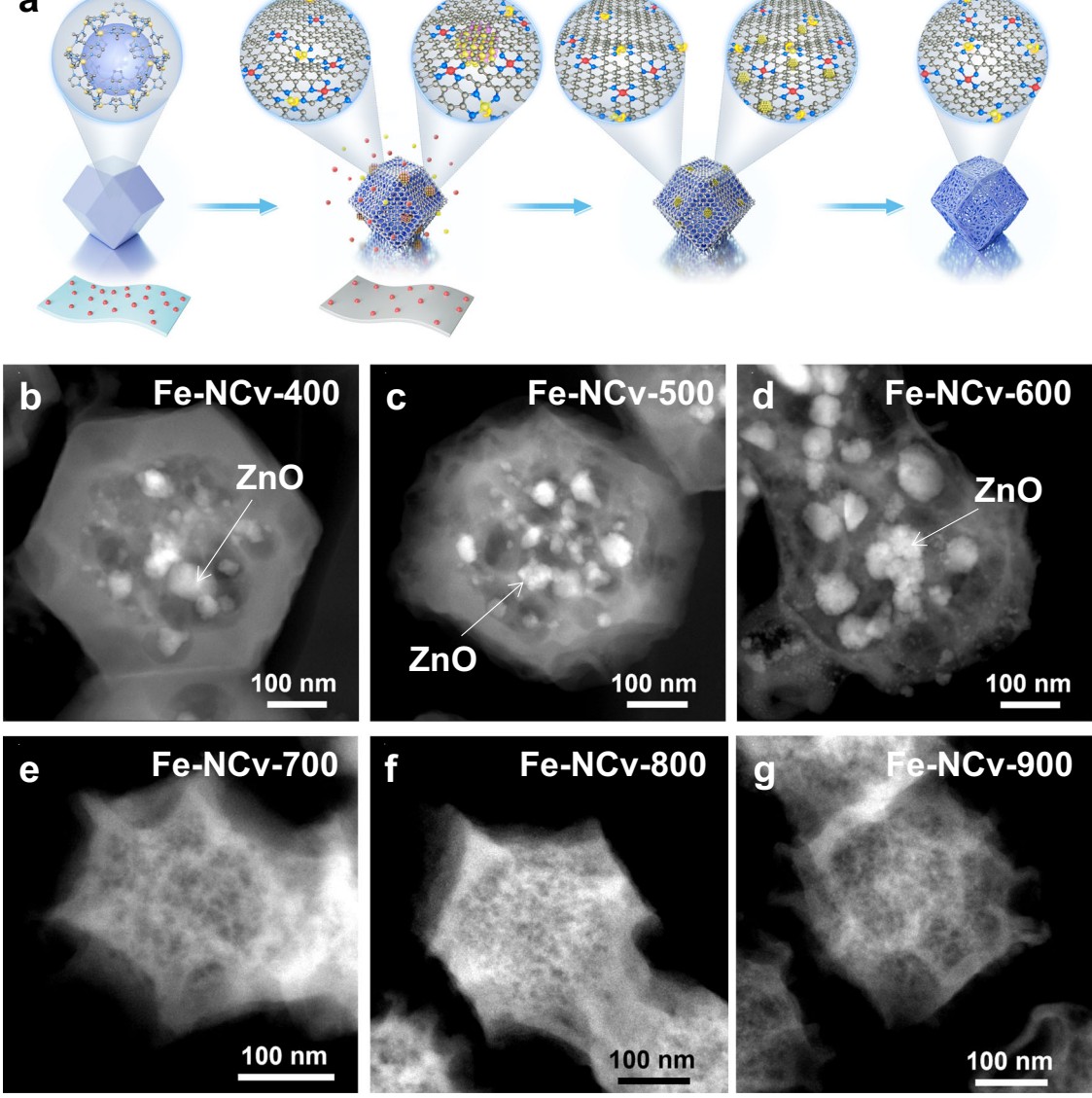

**Fig. 1 | The synthetic procedure of Fe-NCv catalysts by self-carbon-thermal-reduction strategy. a** Schematic illustration of the self-carbon-thermal-reduction strategy. The gray, blue, pink, red and yellow balls represented C, N, O, Fe and Zn atoms, respectively. **b**–**g** HAADF-STEM images of Fe-NCv catalysts after heating from 400 °C to 900 °C for 3 h.

while that of pyridinic N gradually increased, indicating the breaking of Zn-N bonds and formation of coordination-unsaturated N atoms from substrates during agglomeration of $Zn^{2+}$. Above 600 °C, the proportion of graphitic N gradually increased. Nitrogen sorption isotherm experiments were performed to test the Brunauer-Emmett-Teller (BET) surface areas of Fe-NCv catalysts in Supplementary Figs. 20–26. With the in situ formation of ZnO NPs and collapse of ZIF-8, the BET surface areas decreased sharply from 1828 $m^2$/g of Fe-NCv-400, to 991 $m^2$/g of Fe-NCv-500, and to 115 $m^2$/g of Fe-NCv-600. With the in situ disappearance of ZnO NPs during self-carbon-thermal reduction, the BET surface areas increased gradually from 263 $m^2$/g of Fe-NCv-700, to 573 $m^2$/g of Fe-NCv-800, and 853 $m^2$/g of Fe-NCv-900. Besides, mesopores and macropores in situ formed within the CN substrates by self-carbon-thermal reduction. The Raman spectra of Fe-NCv catalysts were exhibited in Supplementary Fig. 27. The two dominant peaks around 1340 $cm^{-1}$ and 1580 $cm^{-1}$ were ascribed to disorder peak (D peak) and graphitic peak (G peak), respectively[45]. The intensities of D peak ($I_D$) and G peak ($I_G$) indicated the degree of disorder and graphitization of carbon substrate, respectively. The $I_D/I_G$ values of Fe-NCv

catalysts increased gradually from Fe-NCv-600 to Fe-NCv-900, with the $I_D/I_G$ values of 0.90, 1.02, 1.05, and 1.07, respectively, indicating the increasing degree of disorder from carbon substrates by gradual formation of carbon-defects. In Supplementary Fig. 28, the electron paramagnetic resonance (EPR) pattern of Fe-NCv-900 exhibited a stronger signal at the g-value of 2.005 g, demonstrating the existence of carbon-vacancies in the carbon substrate of Fe-NCv-900[41].

## XANES and EXAFS measurements of Fe-NCv catalysts

To investigate the coordination environment of Zn and Fe element from Fe-NCv catalysts at atomic level, XANES and EXAFS measurements were performed. In Fig. 3a, the XANES spectra at Zn K-edge of Fe-NCv-400 and Fe-NCv-500 exhibited the similar characteristics compared with ZnO as reference sample. When pyrolysis above 600 °C, the XANES spectra of Fe-NCv catalysts had similar characteristics with zinc phthalocyanine (ZnPc), confirming the in situ disappearance of ZnO by self-carbon-thermal-reaction. Corresponding Fourier-transform EXAFS (FT-EXAFS) spectra in R space were exhibited in Fig. 3b. The two dominant peaks around 1.5 Å and 2.9 Å from ZnO

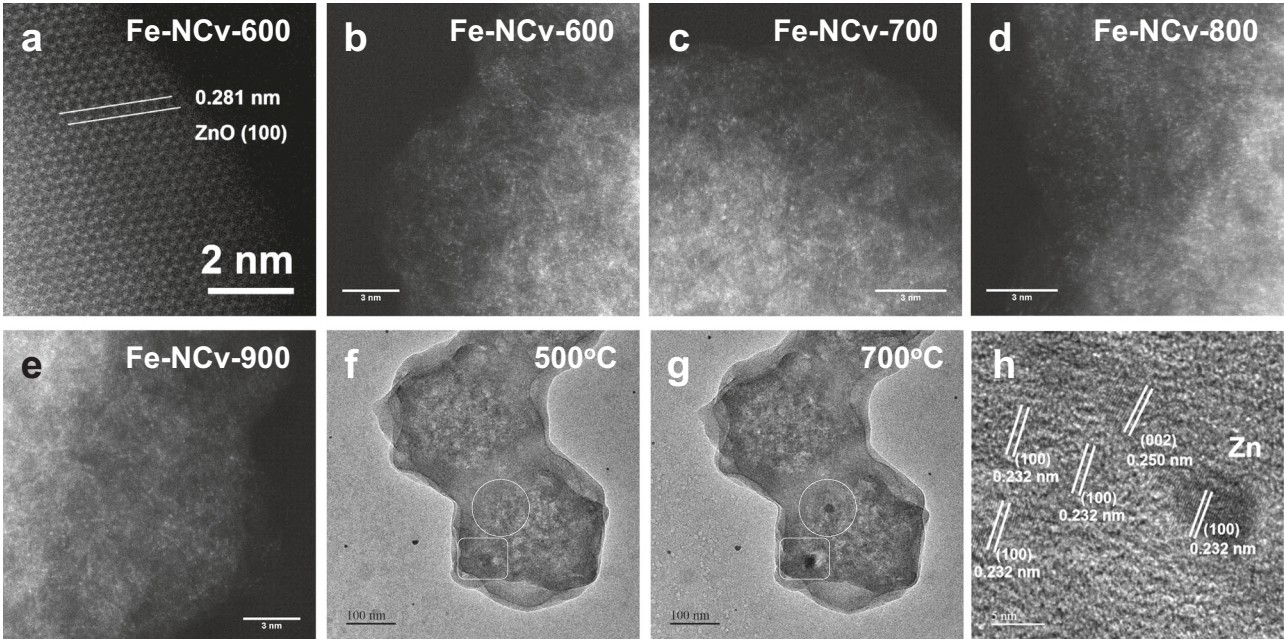

**Fig. 2 | The characterization of Fe-NCv catalysts by AC-STEM and in situ ETEM measurements. a–e** High-resolution AC-STEM images of Fe-NCv catalysts. **f, g** The in situ ETEM images during pyrolysis of Fe-NCv-600 at 500 °C and 700 °C, respectively. **h** The lattice spacing of Zn NPs measured by AC-STEM image.

were ascribed to Zn-O bonds and Zn-O-Zn pathway. From Fe-NCv-400 to Fe-NCv-600, the intensity of Zn-O-Zn peak around 2.9 Å gradually decreased. Above 600 °C, the Zn-O-Zn contribution was negligible from Fe-NCv-700 to Fe-NCv-900, with only one prominent peak around 1.5 Å, similar to the Zn-N bond from ZnPc. In Fig. 3c, the XANES spectra at Fe K-edge from Fe-NCv-400 to Fe-NCv-900 exhibited a similar characteristic with iron phthalocyanine (FePc), especially the characteristic pre-edge peak around 7114 eV, which was assigned to the centrosymmetric Fe-N$_4$ square-planar structure[65]. Corresponding FT-EXAFS spectra in R space were shown in Fig. 3d. The dominant peak around 2.2 Å from Fe foil was assigned to Fe-Fe bonds and the main peak around 1.5 Å from FePc was associated with Fe-N bonds. From Fe-NCv-400 to Fe-NCv-900, only one prominent peak around 1.5 Å existed, similar to that of FePc, without observable Fe-Fe bond around 2.2 Å, demonstrating the Fe element solely existed as Fe ISAS. The wavelet transform (WT) analysis at Zn K-edge was also performed due to its powerful resolutions in both $k$ and R space. From Fe-NCv-400 to Fe-NCv-600, the prominent peak around 6.0 Å$^{-1}$ in $k$ space and 1.5 Å in R space was ascribed to Zn-N/O bonds (Fig. 3e–g). The secondary prominent peak around 4.5 Å$^{-1}$ in $k$ space and 3.2 Å in R space was from Zn-O-Zn pathway of ZnO NPs, with the intensity of Zn-O-Zn peak gradually decreasing from Fe-NCv-400 to Fe-NCv-600. Above 700 °C (Figs. 3h-3j), only one dominant peak (5.0 Å$^{-1}$ in $k$ space and 1.5 Å in R space) existed, which was ascribed to Zn-N bond, with negligible Zn-O-Zn pathway from ZnO, demonstrating the in situ disappearance of ZnO NPs during self-carbon-thermal-reaction. The maximum value of main peak gradually moved towards lower $k$ space, from 6.0 Å$^{-1}$ (Fe-NCv-400, Fe-NCv-500), 5.5 Å$^{-1}$ (Fe-NCv-600), 5.2 Å$^{-1}$ (Fe-NCv-700), 5.0 Å$^{-1}$ (Fe-NCv-800), to 4.8 Å$^{-1}$ (Fe-NCv-900), during the evolution from Zn-N/O coordination into Zn-N coordination, demonstrating the gradually breaking Zn-O bonds. The wavelet transform (WT) analysis at Fe K-edge was shown in Supplementary Fig. 29, confirming the Fe element of the Fe-NCv catalysts existed as isolated single-atom sites.

### The characterization of Fe-NCv-900 and Fe-NC-900 catalysts
As the reference sample, Fe ISAS anchored on N-doped carbon substrate without abundant carbon-defects (Fe-NC-900) was also synthesized by pyrolysis of Fe(acac)$_3$ encapsulated in ZIF-8 at

900 °C. We characterized Fe-NCv-900 and Fe-NC-900 in Fig. 4. The Fe-NCv-900 with defective carbon-substrate had more mesopores and macropores by HAADF-STEM measurement in Fig. 4a. No Fe-based nanoparticles were found in Fe-NCv-900. Corresponding energy dispersive X-ray (EDX) spectroscopy elemental mapping results in Fig. 4b exhibited the carbon, nitrogen and iron element homogeneously dispersed on the Fe-NCv-900. By comparison, the Fe-NC-900 without defective carbon-substrate exhibited no obvious mesopores by HAADF-STEM measurement in Fig. 4c and Supplementary Fig. 30, which was also confirmed by nitrogen sorption isotherm experiments in Supplementary Fig. 31. The carbon, nitrogen, and iron element was evenly distributed on the Fe-NC-900 in Fig. 4d. The Fe content of Fe-NC-900 was 0.78 wt% Fe, similar to that of Fe-NCv-900 with 0.75 wt% Fe, confirmed by ICP-OES measurements. The XPS results and Raman spectrum of Fe-NC-900 were shown in Supplementary Figs. 32 and 33. XANES and EXAFS measurements were performed to test the coordination environment of Fe at atomic level. In Fig. 4e, the positions of near-edge absorption energy of Fe-NCv-900 and Fe-NC-900 located between those of Fe foil and α-Fe$_2$O$_3$, indicating the partial positive charge of Fe element from Fe-NC samples. Corresponding FT-EXAFS spectra in R space of Fe samples were exhibited in Fig. 4f. Both Fe-NCv-900 and Fe-NC-900 had only one prominent peak around 1.5 Å, assigning to Fe-N bonds, without obvious Fe-Fe pathway around 2.2 Å from Fe foil and Fe-O-Fe pathway around 3.0 Å from α-Fe$_2$O$_3$, indicating the Fe element existed as Fe ISAS from Fe-NC samples. The fitting results of Fe-NC samples were exhibited in Fig. 4g, h and Supplementary Table 2, confirming the existence of planar Fe-N$_4$ structure. The WT analysis of Fe samples were also performed in Figs. 4i–l. In Fig. 4i, the prominent peak around 8.0 Å$^{-1}$ in $k$ space and 2.2 Å in R space was ascribed to Fe-Fe bond from Fe foil. In Fig. 4j, the two main peaks (5.0 Å$^{-1}$, 1.5 Å and 8.5 Å$^{-1}$, 3.0 Å) were assigned to Fe-O bond and Fe-O-Fe pathway from α-Fe$_2$O$_3$. By comparison, the WT spectra of Fe-NCv-900 and Fe-NC-900 in Fig. 4k and Fig. 4l only had one dominant peak (4.5 Å$^{-1}$, 1.5 Å), from the Fe-N bonds. Therefore, the atomic dispersion of Fe element from Fe-NCv-900 and Fe-NC-900 was confirmed by XANES, EXAFS measurements, and WT analysis.

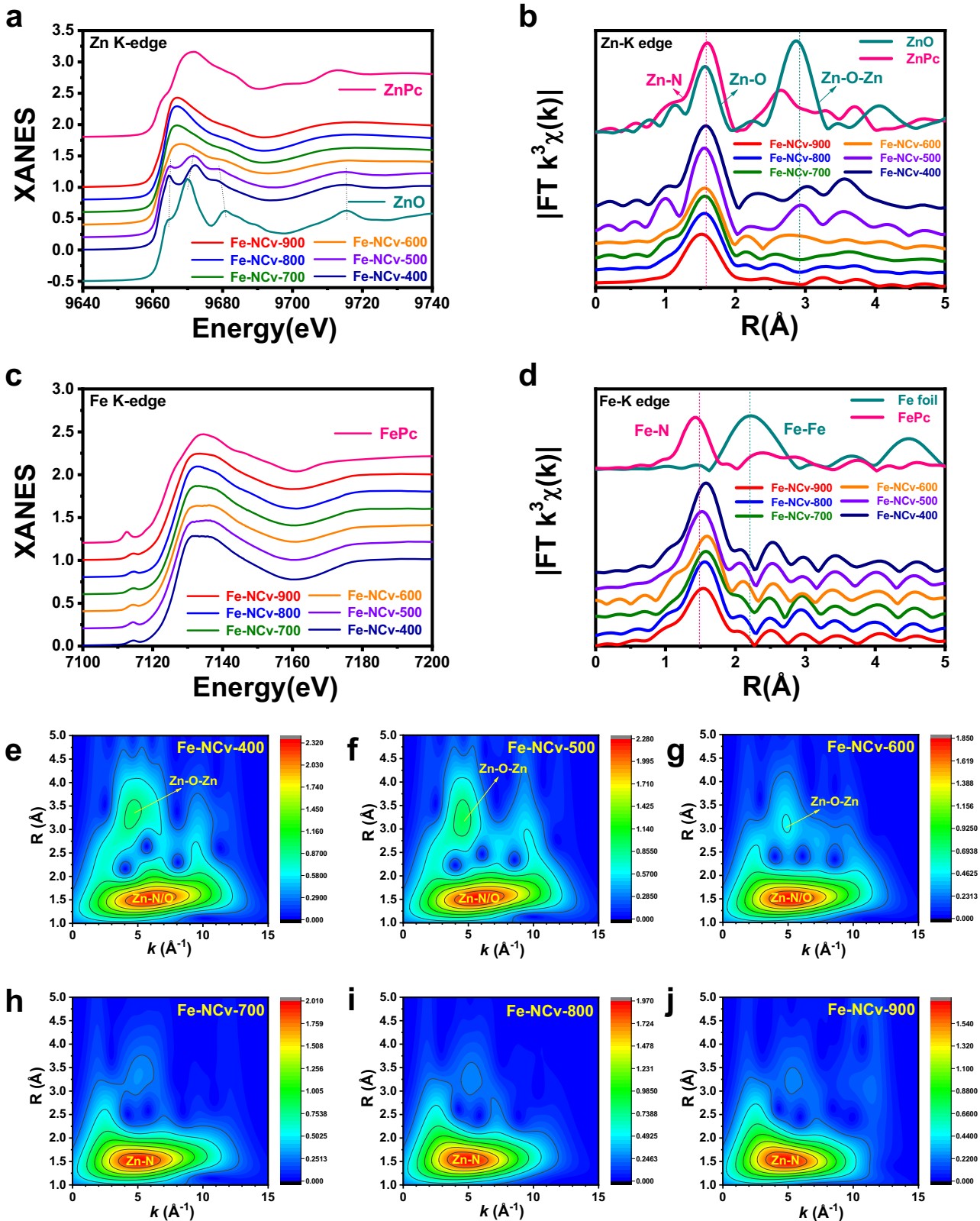

**Fig. 3 | The characterization of Fe-NCv catalysts by XANES and EXAFS measurements. a**, **b** The XANES and FT-EXAFS results in R space of Fe-NCv catalysts at Zn K-edge. **c**, **d** The XANES and FT-EXAFS results in R space of Fe-NCv catalysts at Fe K-edge. **e**–**j** The WT analysis of Fe-NCv catalysts at Zn K-edge.

## Catalytic performance of Fe-NCv catalysts for Fenton-like reactions

Phenol and phenol derivatives are important raw materials in industrial production, such as chemical production, papermaking, pharmaceutical production, and coating industry, but are harmful for human being and environment[66,67]. Efficient degradation of phenolic pollutants in industrial wastewater is significant but with great challenges due to the stability of aromatic rings. Therefore, it is

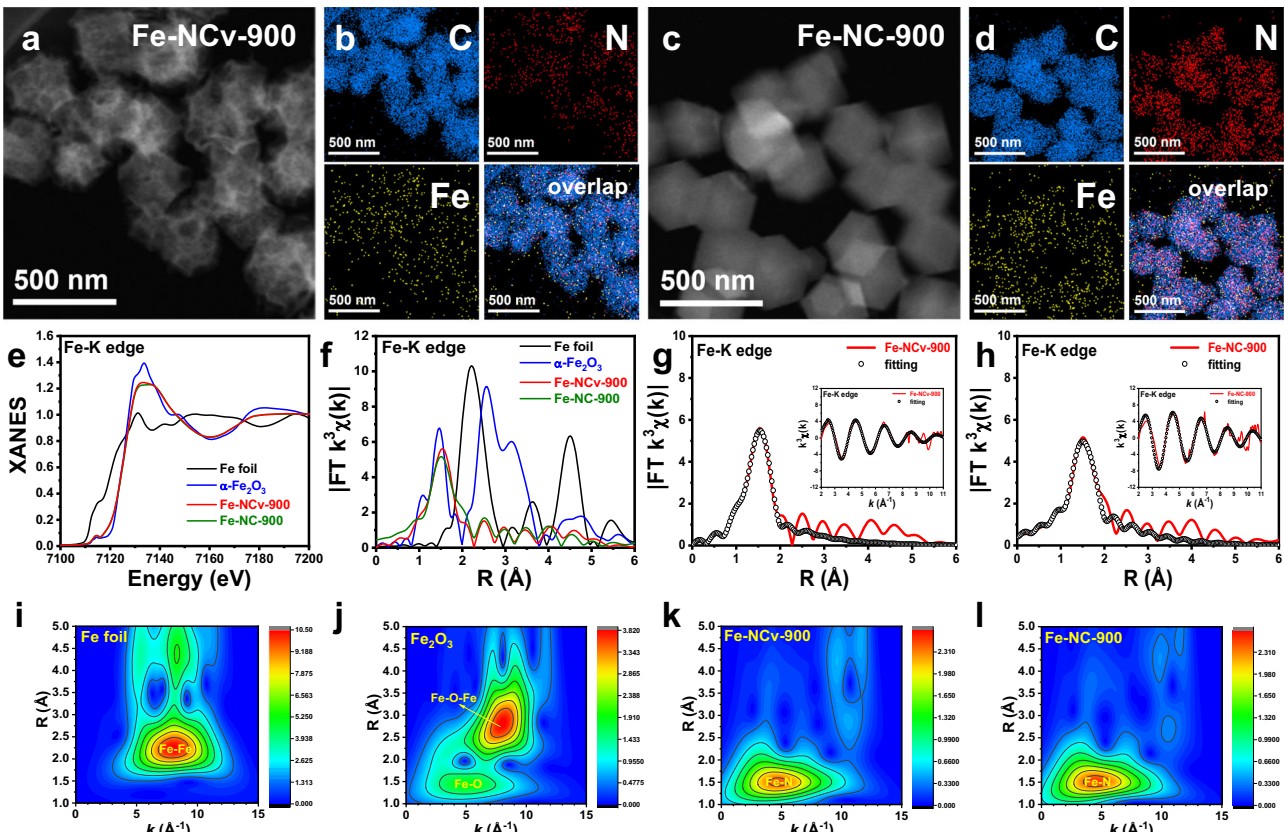

**Fig. 4 | The characterization of Fe-NCv-900 and Fe-NC-900 catalysts.**
**a–d** HAADF-STEM images of Fe-NCv-900, Fe-NC-900, and corresponding EDX spectroscopy elemental mapping results. **e, f** XANES spectra and corresponding FT-EXAFS results in R space of Fe-NCv-900, Fe-NC-900, Fe foil, and α-Fe$_2$O$_3$. **g, h** The fitting results of Fe-NCv-900 and Fe-NC-900 in R space and k space. **i–l** The WT analysis of Fe foil, α-Fe$_2$O$_3$, Fe-NCv-900, and Fe-NC-900, respectively.

meaningful to develop highly efficient catalysts for degradation of phenolic pollutants by Fenton-like reactions.

In Fig. 5, we studied the catalytic performance of Fe-NCv-900 in Fenton-like reactions for degradation of phenol with PMS as oxidant. In Fig. 5a, Fe-NCv-900 exhibited efficient degradation of phenol with the removal ratio of 97% in only 5 min. By contrast, under the same catalytic condition, Fe$_2$O$_3$ and Fe$_3$O$_4$ nano-catalysts as reference samples were basically inert. While the homogeneous 0.075 mg/L Fe$^{3+}$ and Fe$^{2+}$ with the same Fe loading of Fe-NCv-900 exhibited no and low catalytic activities, respectively, demonstrating the Fe ISAS from Fe-NCv-900 were catalytic sites rather than homogeneous Fe ions or Fe-based oxides. Besides, the catalytic performance of NCv-900 without Fe loading was poor with the removal ratio of 10% in 5 min, revealing the rather weak degradation of phenol by NCv substrate. The characterizations of NCv-900, Fe$_2$O$_3$, and Fe$_3$O$_4$ nano-catalysts were shown in Supplementary Figs. 34–40.

In Fig. 5b, we studied the catalytic activities of Fe-NCv-900 for degradation of seven phenolic pollutants. After 5 min, the removal ratios of 4-nitrophenol (4-NP), resorcinol, 4-fluorophenol (4-FP), 4-chlorophenol (4-ClP), 4-bromophenol (4-BrP), 4-Iodophenol (4-IP) and phenol were 56%, 65%, 80%, 87%, 89%, 98%, and 97%, respectively, exhibiting excellent universality for degradation of phenolic pollutants catalyzed by Fe-NCv-900. We compared the first-order kinetic constant k of different phenolic pollutants in Supplementary Fig. 41. With electronegativity of substituent groups on benzene ring decreasing, the removal ratios of phenolic pollutants increased gradually. In Supplementary Fig. 42, we excluded the adsorption of phenolic pollutants on Fe-NCv-900 or the direct degradation by oxidation of PMS.

In Fig. 5c, we compared catalytic activities of Fe-NCv catalysts pyrolysis at different temperatures. As the carbon-defect gradually

formed by self-carbon-thermal-reduction above 600 °C, the catalytic activities of Fe-NCv catalysts increased from Fe-NCv-600 to Fe-NCv-900, with the removal ratios of phenol increasing from 29%, 44%, 69% to 97% within 5 minutes. Fe-NCv-900 was the optimum catalyst with the maximum removal efficiency of phenol. Fe-NCv-1000 was the second best catalyst with removal ratio of phenol of 72% due to the lower atomic utilization efficiency of partial Fe nanoparticles formed at 1000 °C (Supplementary Fig. 43) compared with Fe ISAS from Fe-NCv-900. We compared the first-order kinetic constant k of phenol catalyzed by different Fe-NCv catalysts in Supplementary Fig. 44. We excluded the roles of NCv substrates during catalysis in Supplementary Figs. 45 and 46. The roles of surface area and the structure of the pores on catalytic activity during catalysis were analyzed in Supplementary Figs. 47 and 48.

In Fig. 5d, we compared degradation of phenol between Fe-NC-900 without abundant carbon-defects and Fe-NCv-900 catalysts. Fe-NCv-900/PMS system almost completely degraded phenol within 5 min with removal ratio of 97%, while the Fe-NC-900/PMS system only degraded 27% of phenol. Even after 30 min, 63% phenol remained in Fe-NC-900/PMS system (Fig. 5e). In Fig. 5f, the first-order kinetic constant k for phenol degradation in Fe-NCv-900/PMS system was 0.54 min$^{-1}$, 13.5 times that of Fe-NC-900/PMS system (0.04 min$^{-1}$), indicating Fe-NCv-900 with abundant carbon-defects exhibited enhanced catalytic activity during Fenton-like reactions.

We further investigated the reactive species for Fenton-like reactions. In Supplementary Fig. 49, we explored the reactive oxygen species (ROS) in the Fe-NCv-900/PMS system by EPR experiments with 5,5-dimethyl-1-pyrroline N-oxide (DMPO) as a trapping agent. No signals appeared when PMS or Fe-NCv-900 solely existed while Fe-NCv-900/PMS system exhibited a strong characteristic signal of 5,5-

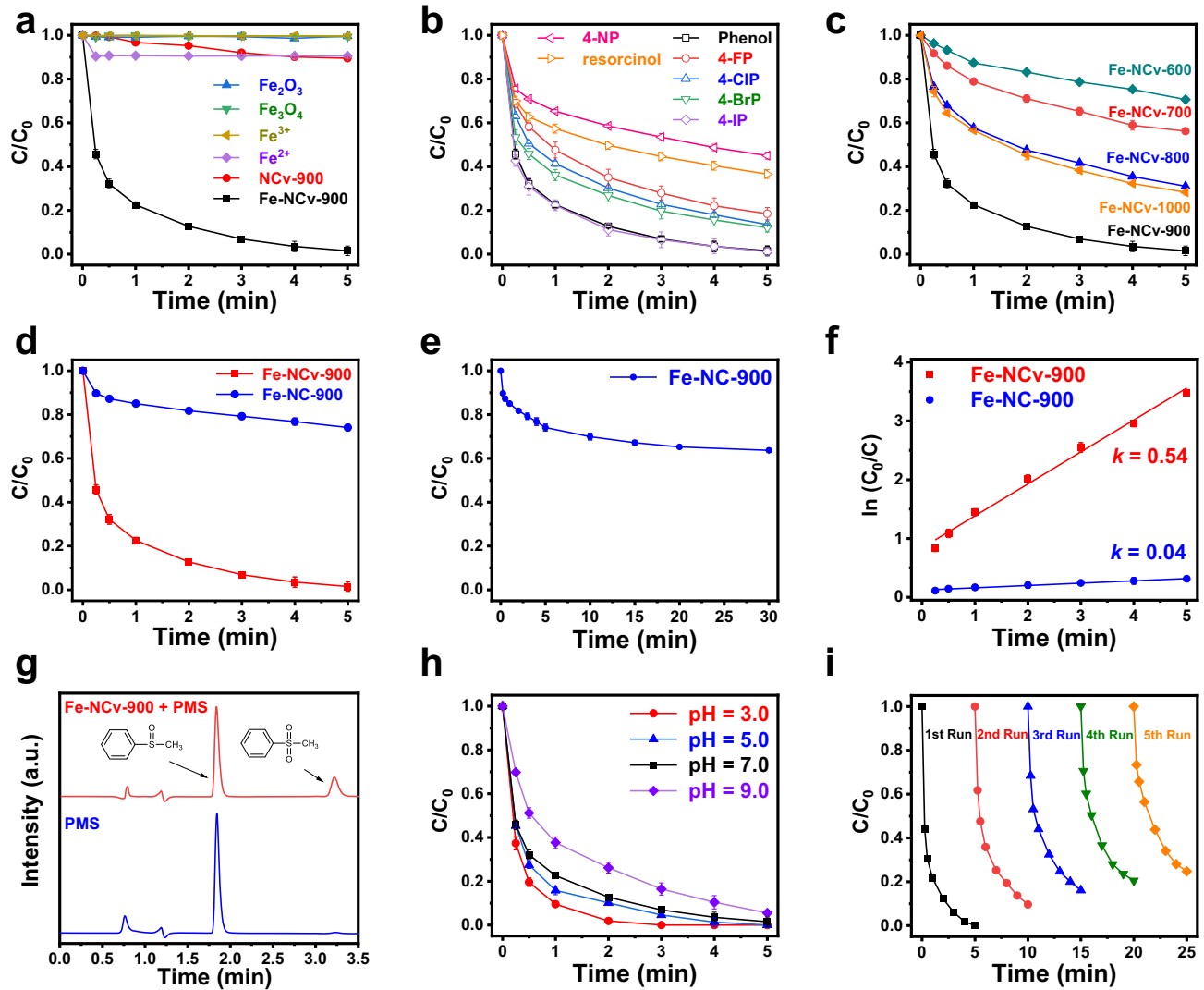

**Fig. 5 | Catalytic performance for Fenton-like reactions. a** Plots of phenol concentration versus time of Fe-NCv-900, NCv-900, $Fe^{2+}$, $Fe^{3+}$, $Fe_2O_3$ and $Fe_3O_4$ as reference samples. (The error bars in Fig. 5 represent SD.) **b** Degradation of 4-NP, resorcinol, 4-FP, 4-ClP, 4-BrP, 4-IP and phenol catalyzed by Fe-NCv-900. **c** Degradation of phenol catalyzed by different Fe-NCv catalysts under pyrolysis at different temperatures. **d**, **e** The comparison of catalytic performance between Fe-NCv-900 and Fe-NC-900 for degradation of phenol. **f** The corresponding plot of ln(C$_0$/C) versus time of Fe-NCv-900 and Fe-NC-900. **g** High performance liquid chromatography (HPLC) spectra with PMSO as a chemical probe of high-valent iron-oxo species. **h** Degradation of phenol catalyzed by Fe-NCv-900 under different pH values. **i** The recyclability tests for degradation of phenol catalyzed by Fe-NCv-900.

dimethyl-1-pyrrolidone-2-oxyl (DMPOX) with peak intensity of 1:2:1:2:1:2:1[58]. Previous study suggested that DMPO could be oxidized into DMPOX by radical or non-radical pathway[58,68–70]. Then we added 100 mM tert-butanol (TBA) or ethanol (EtOH) as quenching agents to scavenge radicals species[58] into Fe-NCv-900/PMS system. The degradation of phenol was not inhibited after adding TBA or EtOH, demonstrating radical was not ROS in this system (Supplementary Fig. 50). Except for radical as ROS, singlet oxygen ($^1O_2$)[71] and high-valent metals species[58,63] as ROS were reported for activation of PMS. When using 2,2,6,6-tetramethylpiperidine (TEMP) as spin-trapping agent for $^1O_2$[58,71], no signals were detected in Supplementary Fig. 51, excluding $^1O_2$ as ROS. In Fig. 5g, methyl phenyl sulfoxide (PMSO) as a chemical probe of high-valent iron-oxo species[58] was added in Fe-NCv-900/PMS system. The peak of methyl phenyl sulfone (PMSO$_2$) by high-performance liquid chromatography (HPLC) measurement demonstrated the formation of high-valent iron-oxo species for activation of PMS. Therefore, we concluded that high-valent iron-oxo species was the ROS in Fe-NCv-900/PMS system, because PMSO was oxidized into PMSO$_2$ by high-valent iron-oxo species. Besides, additional inhibition

experiment in Supplementary Fig. 52 by utilizing Dimethyl sulfoxide (DMSO) as the inhibitor of high-valent iron-oxo species[72] also confirmed the above conclusion. Acetic acid, oxalic acid, lactic acid, and malonic acid as intermediates for degradation of phenol in Fe-NCv-900/PMS system were detected by Gas Chromatography-Mass Spectrometer (GC-MS) measurement, indicating the ring-opening reaction of phenol, consistent with the oxidation pathway by high-valent iron-oxo species[73]. In Supplementary Fig. 53, p-dihydroxybenzene and p-benzoquinone were rapidly degraded within 1 min in Fe-NCv-900/PMS system, indicating p-dihydroxybenzene or p-benzoquinone were not the final products for degradation of phenol. We compared the total organic carbon (TOC) values for degradation of phenol in Supplementary Fig. 54, the decrease of TOC after degradation indicated the deep oxidation of phenol into inorganic carbon. We investigated the degradation of phenol in Fe-NCv-900/PMS system under different pH ranging from 3.0–9.0 in Fig. 5h. After 5 minutes for degradation, the removal ratios of phenol at pH = 3.0, 5.0, 7.0, and 9.0 were 99%, 99%, 97% and 94%, respectively, suggesting the wide-ranging pH universality in Fe-NCv-900/PMS system. We compared the first-order

kinetic constant $k$ of phenol catalyzed by Fe-NCv-900 under different pH in Supplementary Fig. 55. The reusability of Fe-NCv-900 was tested in five consecutive runs in Fig. 5i. The removal ratios of phenol from 1st cycle to 5th cycle catalyzed by Fe-NCv-900 were 97%, 90%, 84%, 80% and 75%, respectively, demonstrating the excellent reusability of Fe-NCv-900. After catalysis, we characterized the used Fe-NCv-900 catalyst, demonstrating the atomic dispersion of Fe element after degradation of phenol in Supplementary Fig. 56. The Fe content of Fe-NCv-900 catalyst after catalysis was 0.72 wt% by ICP-OES measurement, which was similar to the Fe content of Fe-NCv-900 catalyst (0.75 wt%). Except for phenolic compounds, we also investigated the degradation of other organic pollutants, as shown in Supplementary Fig. 57.

## The in situ XANES and EXAFS measurements for Fenton-like reaction

To test the existence form of Fe catalytic sites from Fe-NCv-900 during Fenton-like reaction, in situ XANES and EXAFS measurements were performed. The Fe-NCv-900 catalyst on carbon paper was immersed into 10 μM phenol and 100 μM PMS aqueous solution during in situ Fenton-like reaction. In Fig. 6a, the XANES spectra of Fe-NCv-900 and Fe-NCv-900 during in situ Fenton-like reaction exhibited similar characteristic with that of FePc, with the pre-edge peak around 7114 eV assigned to the centrosymmetric Fe-N$_4$ square-planar structure[65], indicating the Fe element existed as Fe ISAS during catalysis. As shown in the insert of Fig. 6b, the energy of the adsorption edge of Fe-NCv-900 during in situ Fenton-like reaction exhibited an obvious positive shift compared to Fe-NCv-900. Besides, the white line intensity around 7135 eV of Fe-NCv-900 during in situ Fenton-like reaction was higher than that of Fe-NCv-900, confirming the formation of high-valent iron-oxo species[58]. The corresponding Fourier-transform EXAFS (FT-EXAFS) spectra in R space were shown in Fig. 6c. During in situ Fenton-like reaction, the intensity of the dominant peak of Fe-NCv-900 at around 1.5 Å obviously increased compared to Fe-NCv-900 catalyst, which was assigned to the in situ formation of Fe-O bond from high-valent iron-oxo species. Compared with the WT contour plot of Fe-NCv-900 in Fig. 6d, the intensity of the dominant peak (4.5 Å$^{-1}$ in $k$ space, 1.5 Å in R space) of Fe-NCv-900 during in situ Fenton-like reaction in Fig. 6e obviously increased and the spatial distribution of the dominant peak during in situ Fenton-like reaction moved towards higher $k$ space, owing to the formation of Fe-O bonds from high-valent iron-oxo species during catalysis, which was similar to the Fe-O pathway from Fe$_2$O$_3$ in Fig. 6f. Compared with Fe$_2$O$_3$, the absence of Fe-O-Fe pathway (8.5 Å$^{-1}$, 3.0 Å) from Fe-NCv-900 during in situ Fenton-like reaction demonstrated the sole existence of Fe-ISAS, revealing the excellent stability of Fe-ISAS from Fe-NCv-900 during catalysis.

## DFT calculation for degradation of phenol by Fenton-like reaction

In Fig. 7, we studied the effect of carbon-atom defects on the catalytic performance of Fe-N$_4$ sites on N-doped carbon for degradation of phenol by DFT calculation. We constructed Fe-N$_4$ site and three defective Fe-N$_4$ sites with a carbon-atom vacancy on N-doped carbon substrate in Fig. 7a. The formation energy of carbon-atom vacancy on pristine graphene and Fe-N$_4$-sites-doped graphene were compared in Supplementary Fig. 58. The formation energy of carbon-atom vacancy on pristine graphene was 7.82 eV. By comparison, carbon-atom vacancy could be easily formed around Fe-N$_4$-sites, such as Fe-N$_4$/Cv-1, Fe-N$_4$/Cv-2, and Fe-N$_4$/Cv-3, with formation energy of 5.49 eV, 5.92 eV, and 7.55 eV, respectively. Compared with Fe-N$_4$/Cv-3, carbon-atom vacancies directly adjacent to Fe-N$_4$ sites were more stable with lower formation energies, such as Fe-N$_4$/Cv-1 and Fe-N$_4$/Cv-2.

To study the formation mechanism of carbon-atom vacancies during self-carbon-thermal-reduction between carbon-substrates and ZnO, we constructed the ZnO@graphene and ZnO@Fe-N$_4$/N-doped

graphene hetero-structures in Supplementary Figs. 59-62. The formation of C-O bond between carbon-substrate and ZnO was the key step during formation of carbon-vacancy defects. The top view and side view of the optimized structures of ZnO@graphene and ZnO@Fe-N$_4$/N-doped graphene hetero-structures before and after C-O bonding were exhibited in Fig. 7b, and Supplementary Figs. 59–62. In Fig. 7c, the energy change for C-O bonding of ZnO@graphene hetero-structure was 3.40 eV, higher than those of ZnO@Fe-N$_4$/Cv-1 (2.41 eV), ZnO@Fe-N$_4$/Cv-2 (2.56 eV) and ZnO@Fe-N$_4$/Cv-3 (2.69 eV), indicating the easier formation of C-O bonding adjacent to Fe-N$_4$ sites.

We studied the catalytic pathway on Fe-N$_4$ sites/N-doped carbon for degradation of phenol in Fig. 7d and Fig. 7e. As we revealed that high-valent iron-oxo species were the ROS during catalysis, axial O atom coordinated with Fe-N$_4$ sites (O-Fe-N$_4$) served as reactive intermediate. As shown in Fig. 7e, five steps were involved for degradation of phenol: the formation of O-Fe-N$_4$ sites (S1) from Fe-N$_4$ sites (S0), adsorption of phenol on O-Fe-N$_4$ sites (S1 to S2), oxidation of phenol by formation of C-O bond (S2 to S3), hydrogen-atom transfer from phenol to O-Fe-N$_4$ sites (S3 to S4) and desorption of product (S4 to S5). There were three possible oxidation sites of phenol during catalysis: ortho position (o-site), meta position (m-site) and para position (p-site) of hydroxyl group. We compared the selectivity of different oxidation sites for oxidation of phenol in Supplementary Fig. 63. There was no trend on m-site for oxidation of phenol, while the rate-determining steps (RDS) on o-site and p-site were transformation from S2 to S3. Oxidation on p-site had lower energy barrier of 1.03 eV for RDS compared with o-site (1.28 eV), indicating more easier oxidation of phenol on p-site rather than o-site.

Then we studied the effect of carbon-atom vacancy on the catalytic activity of Fe-N$_4$ sites for oxidation of phenol. In Fig. 7d, the RDS for pristine Fe-N$_4$/N-doped graphene without carbon-vacancy was oxidation of phenol by formation of C-O bond (S2 to S3) with energy barrier of 1.03 eV. While the RDS for Fe-N$_4$/Cv-1 and Fe-N$_4$/Cv-2 were desorption of product (S4 to S5) with energy barriers of 0.69 eV and 0.68 eV, respectively. The RDS for Fe-N$_4$/Cv-3 was oxidation of phenol by formation of C-O bond (S2 to S3) with energy barrier of 0.80 eV. Compared with pristine Fe-N$_4$ site without carbon-atom vacancy, Fe-N$_4$/Cv-1, Fe-N$_4$/Cv-2, and Fe-N$_4$/Cv-3 had much lower energy barriers for RDS, indicating the adjacent carbon-atom vacancy enhanced the catalytic activity of Fe-N$_4$ sites for degradation of phenol, consistent with experimental results that Fe-NCv-900 exhibited much better catalytic activity than Fe-NC-900 during Fenton-like reaction. Besides, the energy barriers during formation of C-O bond (S2 to S3) on pristine Fe-N$_4$, Fe-N$_4$/Cv-1, Fe-N$_4$/Cv-2, and Fe-N$_4$/Cv-3 sites were 1.03 eV, 0.51 eV, 0.28 eV, and 0.80 eV, respectively, demonstrating the easier oxidation of phenol and easier formation of C-O bonds catalyzed by Fe-N$_4$ sites with adjacent carbon-atom vacancy.

We compared the chemical structures and electronic distributions between pristine Fe-N$_4$, Fe-N$_4$/Cv-1, Fe-N$_4$/Cv-2, and Fe-N$_4$/Cv-3 in Supplementary Table 3 and Supplementary Fig. 64. Fe-N$_4$/Cv-1, and Fe-N$_4$/Cv-2 with carbon-atom vacancy directly adjacent to Fe-N$_4$ sites, had shorter Fe-N bond length, higher valence states of Fe atoms and lower positions of d-band center, compared with pristine Fe-N$_4$ site. While Fe-N$_4$/Cv-3 with carbon-atom vacancy non-directly adjacent to Fe-N$_4$ sites had similar properties to pristine Fe-N$_4$ site.

We analyzed O-Fe-N$_4$ sites (S1) in Supplementary Table 4 and Supplementary Fig. 65. Compared with pristine Fe-N$_4$, introducing carbon-atom vacancies adjacent to Fe-N$_4$ sites increased the net Bader charge of Fe atoms with higher valence states and the axial O atoms obtained more electrons (Supplementary Table 4). The integrated crystal orbital Hamilton population (ICOHP) absolute values of Fe-O bonds from Fe-N$_4$/Cv-1, Fe-N$_4$/Cv-2, and Fe-N$_4$/Cv-3 were lower than that of pristine Fe-N$_4$ site, indicating carbon-atom vacancies adjacent to Fe-N$_4$ sites weakened interaction of Fe-O bonds and facilitated the oxidation of phenol during C-O bonding.

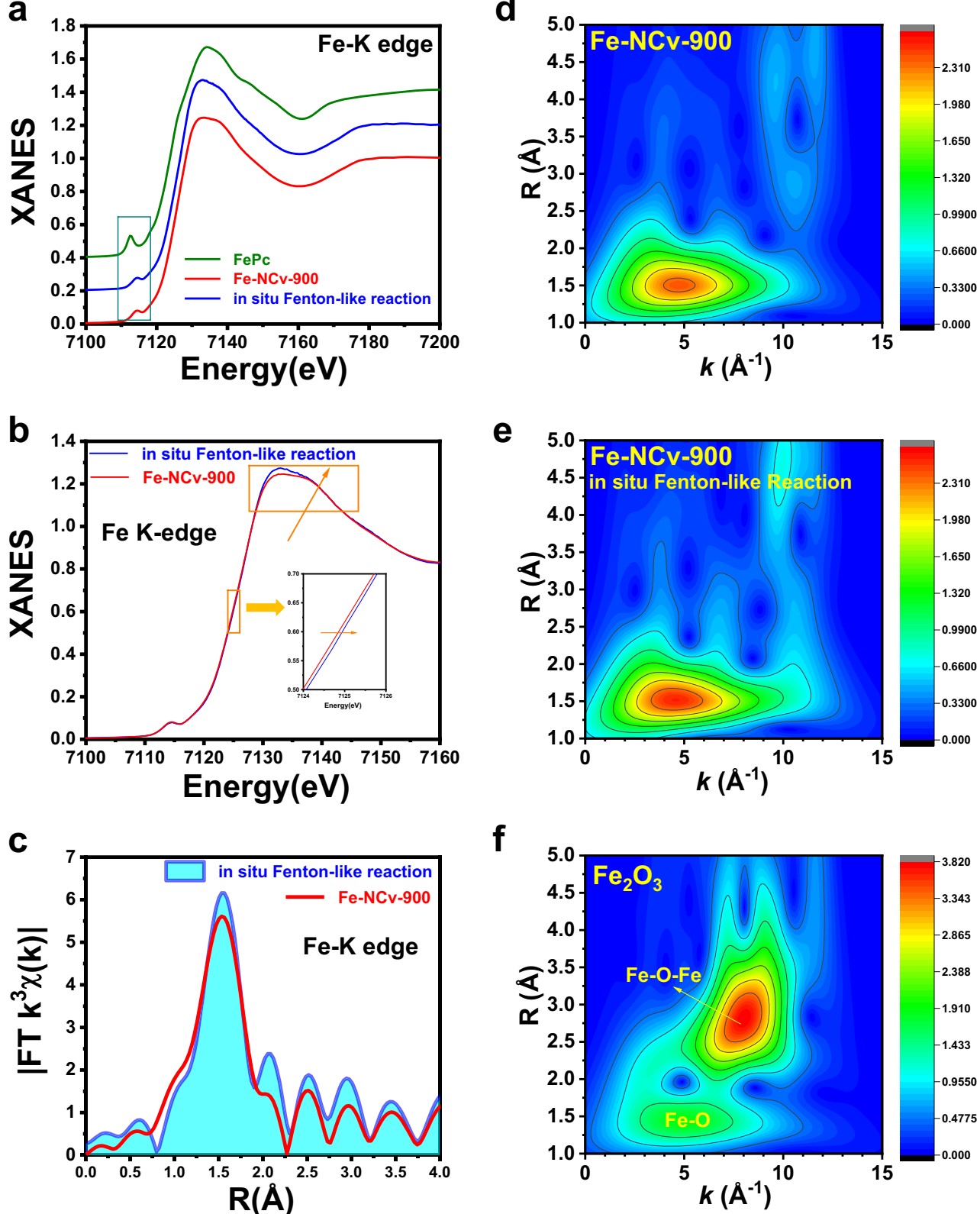

**Fig. 6 | The in situ XANES and EXAFS measurements for Fenton-like reaction.**
**a** The XANES spectra of FePc, Fe-NCv-900, and Fe-NCv-900 during in situ Fenton-like reaction. **b** The comparison of XANES spectra of Fe-NCv-900 and Fe-NCv-900 during in situ Fenton-like reaction. **c** The corresponding FT-EXAFS spectra in R space of Fe-NCv-900 and Fe-NCv-900 during in situ Fenton-like reaction. **d–f** The WT analysis of Fe-NCv-900, Fe-NCv-900 during in situ Fenton-like reaction and Fe$_2$O$_3$.

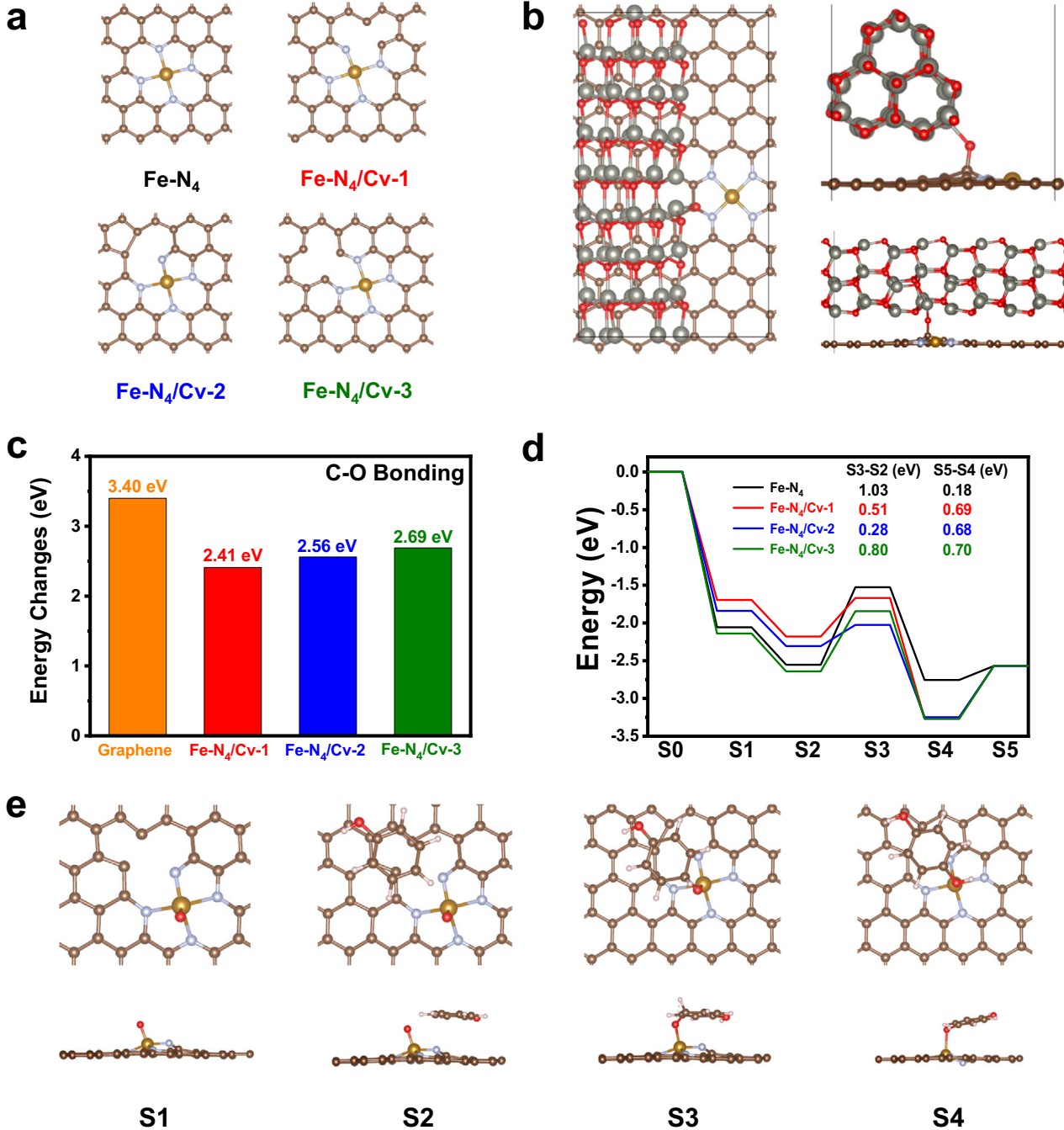

**Fig. 7 | DFT calculation for degradation of phenol by Fenton-like reaction. a** The optimized structures of pristine Fe-N₄ site and three defective Fe-N₄ sites with a carbon-atom vacancy on N-doped carbon substrate. **b** The top view and side view of the optimized structure of ZnO@Fe-N₄/N-doped graphene hetero-structure for synthesis of Fe-N₄/Cv-1 after C-O bonding. **c** The comparison of energy changes for C-O bonding of ZnO@graphene, ZnO@Fe-N₄/N-doped graphene hetero-structures. **d** The catalytic pathways for degradation of phenol catalyzed by different Fe-N₄ sites/N-doped carbon and corresponding energy changes. **e** The corresponding optimized structures of S1, S2, S3, and S4.

We compared the interaction of Fe-O bonds and C-O bonds in S3 models by ICOHP analysis in Supplementary Table 5. After introducing carbon-atom vacancies adjacent to Fe-N₄ sites, the ICOHP absolute values of Fe-O decreased while those of C-O bonds increased, indicating carbon-atom vacancies adjacent to Fe-N₄ sites weakened Fe-O bonds and strengthened C-O bonds, which facilitated the oxidation of phenol. Compared with pristine Fe-N₄ site, carbon-atom vacancies increased the Fe-O bond length and shortened the C-O bond length. By Bader charge analysis on S3 models in Supplementary Fig. 66, more electrons were transferred from C atom to O atom of C-O bond after

introducing carbon-atom vacancies. Therefore, DFT calculation revealed the adjacent carbon-atom vacancy of Fe-N₄ sites facilitated the C-O bonding and lowered the energy barrier of RDS during oxidation of phenol by Fenton-like reaction.

## Discussion

In summary, this work skillfully constructed carbon-defect engineering on single Fe-N₄ sites in ZnO-Carbon nano-reactor by self-carbon-thermal-reduction strategy. The Fe-NCv-900 exhibited much higher activity, excellent recyclability, and wide pH suitability for Fenton-like

reaction, with the first-order rate constant sharply increasing by 13.5 times compared with Fe-NC-900. EXAFS and in situ ETEM measurements revealed the formation mechanism of carbon-defects by self-carbon-thermal-reduction strategy. DFT calculation demonstrated the adjacent carbon-vacancies efficiently boosted the catalytic activity of Fe-N$_4$ sites for degradation of phenol. This work provides a rational direction for designing carbon-defect engineering on metal ISAS catalyst by self-carbon-thermal-reduction strategy, improving the activity during catalysis, studying the formation mechanism of carbon-defects, and revealing the relationship between carbon-defect and catalytic activity.

## Methods

### Reagents

zinc nitrate hexahydrate (Zn(NO$_3$)$_2$·6H$_2$O, Aladdin, 99.99%), 2-methylimidazole (Acros, 99%), methanol (Beijing Chemical Reagent), ethanol (Beijing Chemical Reagent), iron(III) acetylacetonate (Fe(acac)$_3$, Acros, 99 + %). 5,5-Dimethyl-1-pyrroline-N-oxide (DMPO) and PMS (in the form of Oxone, KHSO$_5$ ≥ 42.80%) were obtained from Sigma-Aldrich Chemical Co., Ltd. Fe$_2$O$_3$, Fe$_3$O$_4$, FeSO$_4$, Fe$_2$(SO$_4$)$_3$, phenol, 4-nitrophenol (4-NP), resorcinol, 4-fluorophenol (4-FP), 4-chlorophenol (4-ClP), 4-bromophenol (4-BrP), 4-iodophenol (4-IP), p-dihydroxybenzene, p-benzoquinone, tert-butanol (TBA), ethanol (EtOH), methanol (MeOH, HPLC), acetonitrile (HPLC), 2,2,6,6-tetra-methylpiperidine (TEMP), methyl phenyl sulfoxide (PMSO), methyl phenyl sulfone (PMSO$_2$), aniline, tetracycline (TC), p-Hydroxybenzoic acid (PHBA), bisphenol A (BPA), sulfamethoxazole (SMX) and dimethyl sulfoxide (DMSO) were purchased from Shanghai Aladdin Biochemical Technology Co., Ltd. All chemicals were of analytical grade and used as received. The water during experiments was ultrapure (Millipore Milli-Q grade) with a resistivity of 18.2 MΩ.

### Preparation of ZIF-8

For synthesis of ZIF-8, 3 g Zn(NO$_3$)$_2$·6H$_2$O was dissolved in 40 ml methanol by ultrasonic treatment for 20 minutes. Then 6.5 g 2-methylimidazole was dissolved in 80 ml methanol by ultrasonic treatment for 20 minutes. Next, the above 2-methylimidazole solution and the above Zn(NO$_3$)$_2$·6H$_2$O solution were fixed together under room temperature around 25 °C, and were stirred violently for 24 h. After crystallization of ZIF-8, we collected ZIF-8 powder by centrifugation at about 2,3870 x g for 5 minutes, washed with methanol for three times and dried at 80 °C for 5 h.

### Preparation of Fe(acac)$_3$@filter papers

We referred to a reported ref. 74 for preparing Fe(acac)$_3$@filter papers with modifications in synthetic method. We dissolved 800 mg Fe(acac)$_3$ in 20 ml ethanol. The above Fe(acac)$_3$/ethanol mixture was treated by ultrasonic treatment for 30 minutes in order to prepare 40 mg Fe(acac)$_3$/ml ethanol solution. Subsequently, we stirred the above Fe(acac)$_3$/ethanol solution at room temperature. The rectangular filter papers (2 cm×1.5 cm) were immersed into the above 40 mg Fe(acac)$_3$/ml ethanol solution by stirring for several seconds to absorb the Fe(acac)$_3$ molecules into the rectangular filter papers. The redundant Fe(acac)$_3$ ethanol solution on the surface of filter papers was removed before drying. We put the Fe(acac)$_3$@filter papers in an oven at 80 °C for several minutes in order to evaporate ethanol.

### Preparation of Fe-NCv-900 catalysts

We synthesized Fe-NCv-900 catalyst by self-carbon-thermal-reduction. We referred a reported ref. 74 during synthesis. As shown in Supplementary Fig. 1a, the bottom of a ceramic boat was evenly paved with eight Fe(acac)$_3$@filter papers (2 cm × 1.5 cm). Then, the above eight Fe(acac)$_3$@filter papers were homogeneously paved with 200 mg ZIF-8 powder, as shown in Supplementary Fig. 1b. Subsequently, the ZIF-8 powder was evenly paved with other eight Fe(acac)$_3$@filter papers (2 cm×1.5 cm) to form the sandwich-like structure, as shown in Supplementary Fig. 1c. Finally, the precursors in the ceramic boat were placed in a tube furnace. We heated the sample to 900 °C and the heating rate was 5 °C/min. After pyrolysis at 900 °C for 3 h under flowing argon gas, the sample was cooled to room temperature naturally. As shown in Supplementary Fig. 1d, the Fe-NCv-900 catalyst was obtained after easy separation between the Fe-NCv-900 catalyst and the carbonized filter papers. Similarly, the Fe-NCv-400, Fe-NCv-500, Fe-NCv-600, Fe-NCv-700, Fe-NCv-800 and Fe-NCv-1000 catalysts were synthesized by a similar synthetic method, except for pyrolysis at 400 °C, 500 °C, 600 °C, 700 °C, 800 °C and 1000 °C for 3 h, respectively. For synthesis of NCv-900, NCv-800, NCv-700, and NCv-600 without Fe loading, the synthetic method was similar to those of Fe-NCv catalysts, except for utilization of ordinary filter papers rather than Fe(acac)$_3$@filter papers.

### Preparation of Fe-NC-900 catalyst

We synthesized Fe-NC-900 by pyrolysis of Fe(acac)$_3$@ZIF-8 composite. For synthesis of Fe(acac)$_3$@ZIF-8, 3 g Zn(NO$_3$)$_2$·6H$_2$O and 1104 mg Fe(acac)$_3$ were dissolved in 40 ml methanol by ultrasonic treatment for 20 minutes. Then 6.5 g 2-methylimidazole were dissolved in 80 ml methanol by ultrasonic treatment for 20 minutes. Next, the above 2-methylimidazole solution and the above Fe(acac)$_3$-Zn(NO$_3$)$_2$·6H$_2$O solution were fixed together under room temperature around 25 °C, and were stirred violently for 24 h. After crystallization of Fe(acac)$_3$@ZIF-8, we collected Fe(acac)$_3$@ZIF-8 powder by centrifugation at about 23,870×g for 5 min, washed with methanol for six times and dried at 80 °C for 5 h. For synthesis of Fe-NC-900, we heated the Fe(acac)$_3$@ZIF-8 to 900 °C with a heating rate of 5 °C/min and we kept the temperature for 3 h under flowing argon gas and then the Fe-NC-900 sample was cooled naturally to room temperature.

### Characterization

We utilized the Rigaku RU-200b X-ray powder (XRD) diffractometer with Cu Kα radiation ($\lambda$ = 1.5418 Å) to measure the crystalline structure and phase purity of Fe-NCv samples. Inductively coupled plasma optical emission spectrometry (ICP-OES) was performed to determine the exact Fe and Zn contents of Fe-NCv samples. The high-resolution HAADF-STEM images and corresponding elemental mapping results of Fe-NCv samples were measured by a JEOL-2100F FETEM with electron acceleration energy of 200 kV. We observed the atomic structures of Fe-NCv samples by AC-STEM with an ARM-200CF (JEOL, Tokyo, Japan) transmission electron microscope operated at 200 keV and equipped with double spherical aberration (Cs) correctors. The attainable resolution of the probe defined by the objective pre-field was 78 picometers. We performed the X-ray photoelectron spectroscopy (XPS) measurement of Fe-NCv samples on ESCALAB 250 Xi X-ray photoelectron spectrometer with Al Kα radiation. The binding energies of Fe-NCv samples were calibrated by setting the measured binding energy of C 1$s$ to 284.8 eV. We switched on an electron flood gun to prevent the surface charging of Fe-NCv samples during XPS measurement. We carried out the nitrogen sorption isotherm experiments of Fe-NCv samples by the method of Brunauer-Emmett-Teller (BET) with a QuadraSorb SI automated surface area and pore size analyzer (Quantachrome Instruments) at 77 K. The Fe-NCv samples were degassed at 200 °C before measurement. Barrett-Joyner-Halenda (BJH) method was utilized to analyze the distribution of mesopores and micropores. The thermogravimetric analysis coupled with fourier-transform infrared spectroscopy and mass spectrometry (TG-FTIR-MS) measurement of Fe(acac)$_3$@filter papers was performed on a STA449F3 Synchronous Thermal Analyzer (NETZSCH, Germany), a QMS 403 C Mass Spectrometer (NETZSCH, Germany) and a VERTEX 70 V Infrared Spectrometer (BRUKER, Germany). The temperatures of

the detector and transmission part were 200 °C. The vacancy defect structures of Fe-NCv samples were determined by electron paramagnetic resonance (EPR, BRUKE EMXPLUS) measurement.

### In situ ETEM measurement

The in situ ETEM measurements were performed on a Titan ETEM microscope (FEI) (300 kV with an image Cs-corrector). We diluted the Fe-NCv-600 sample in isopropanol and dispersed the Fe-NCv-600 sample by ultrasonic. Subsequently, the Fe-NCv-600 /isopropanol solution was cast on a MEMS in situ heating chip. The in situ heating chip with Fe-NCv-600 sample was blow dried by dry air and was mounted on a functional TEM holder (DENSsolutions, wildfire). For avoiding cross pollution, both the chip and the sample holder were treated by plasma clean for 30 minutes. After above treatment, the chip and the holder were further cleaned by UV for 5 minutes under vacuum in order to remove the residual organic pollutants. The gaseous environment of the ETEM column was pure Ar gas (99.9995%). We heated the Fe-NCv-600 sample by the MEMS heater, which was integrated on the chip.

### EXAFS measurement

The spectra of X-ray absorption fine structure at Fe K-edge and Zn K-edge were collected at the TPS-21A beamline of the National Synchrotron Radiation Research Center (NSRRC, Hsinchu, Taiwan, operated at 3 GeV with a maximum current of 500 mA). The EXAFS measurement was performed at room temperature in fluorescence mode by utilization of a Lytle detector. The powder of catalysts were pelletized as a disks (diameter of 8 mm) with polyvinylidene fluoride (PVDF) powder as binders. The energies of spectra at Fe K-edge and Zn K-edge were corrected by the absorption edges of Fe foil and Zn foil, respectively. For in situ XANES and EXAFS measurements, the Fe-NCv-900 catalyst on carbon paper was immersed into 10 μM phenol and 100 μM PMS aqueous solution during in situ Fenton-like reaction. The fluorescence mold was performed for in situ XANES and EXAFS measurements.

### EXAFS data analysis

The XAFS data normalization and Fourier-transformed data fitting were performed by using Demeter (version 0.9.26) software package. During fitting, $k^3$ weights, k-range (3- -11 Å$^{-1}$), and R range (1--4 Å) were applied for all of the data.

The $\chi(k)$ data format was imported into the Larch Python code for Wavelet Transform analysis. The parameters were listed as follows: R range was 1-5 Å; k range was 2–14 Å$^{-1}$; k-weight was 3; and the Morlet function with $\kappa = 6$, $\sigma = 1$ was utilized as the mother wavelet to provide the overall distribution.

### Catalytic measurement for Fenton-like reaction

For Fenton-like reaction, 0.80 mg Fe-NCv catalysts were added into 80 mL 10 μM contaminant aqueous solution in a 100 mL beaker under stirring at 600 rpm. Subsequently, 80 μL PMS (100 mM) was injected to the beaker to initiate the reaction with solution pH adjusted by HClO₄ or NaOH solution at 25 °C. Buffers were not utilized during catalysis. At different reaction time, 1.0 mL solution from catalytic system was removed and filtered by a 0.22 μm PTFE filter, followed by immediate quenching with excessive Na₂SO₃ and EtOH before testing the concentration of residue contaminants in the catalytic system. We performed the batch experiments in at least duplicate and the results presented are average values with standard deviations.

### Density functional theory calculation

We performed the DFT calculations with VASP package, utilizing GGA-PBE functional and projector-augmented wave (PAW) potentials to account for core-valence interactions. We carried out spin-polarized Kohn-Sham calculations with a kinetic energy cutoff of 400 eV for plane wave expansions. We optimized all the geometries with self-consistent field energy and force criteria set to be $10^{-6}$ eV and 0.02 eV Å$^{-1}$. We adopted Grimme's DFT-D3 scheme to consider van der Waals interactions. We calculated the electronic states and charge transfer through Bader charge analysis. We performed the crystal orbital Hamiltonian population (COHP) calculations with Lobster package to give a fundamental, energy-resolved understanding of bonding.

We constructed two models to study the formation of C vacancies and the reaction pathway of phenol oxidation by Fenton-like reaction, respectively. For ZnO@Fe-N₄/N-doped graphene hetero-structures, the atomic positions and lattice parameters of a 3-layer ZnO (100) slab on pure graphene was optimized with the z-axis length fixed. The reciprocal space was sampled by the Γ-centered Monkhorst-Pack scheme with a grid of $2 \times 1 \times 1$. The optimized lattice parameters were as follows: $a = 12.45$ Å, $b = 21.26$ Å, $c = 30.00$ Å, $\alpha = \beta = \gamma = 90^\circ$. The shortest distance from graphene plane to ZnO (100) surface was 2.90 Å. To facilitate computation, we truncated the ZnO slab into a nanotube-like structure with three (100) surfaces. We constructed FeN₄ sites on N-doped graphene after replacing 2 central C atoms with one Fe atom and changing neighboring 4 C atoms to N atoms on graphene. Then, the formation of the C-O bond could be simulated by moving the O atom in the ZnO lattice to bond with C on the graphene. The energy differences before and after C-O bonding were used to compare the tendency of formation for different types of C vacancies around Fe-N₄ sites.

Other models were based on pristine graphene's hexagonal supercell ($6 \times 6$ graphene unit cells). We utilized a vacuum layer of 20 Å along the z direction to eliminate interlayer interactions. The K-mesh used in this model was $3 \times 3 \times 1$. To investigate the effect of C vacancies on Fe-N₄ sites during catalysis, we removed different adjacent C atoms to Fe-N₄ sites. Specifically, single C vacancies with different locations were examined in this study to avoid large distortion and instability. We calculated the formation energy of C vacancy as follows:

$$\Delta E = E(\text{substrate with 1C vacancy}) + \frac{1}{2}E(\text{graphene unit cell}) \\ - E(\text{substrate with no C vacancy})$$

For the reaction pathways, Fe-N₄ sites were oxidized into O-Fe-N₄ sites by H₂SO₅, firstly. The energy difference between S0 and S1 was calculated as follows:

$$\Delta E_{S0-S1} = E(Fe - N_4) + E(H_2SO_5) - E(O - Fe - N_4) - E(H_2SO_4)$$

O-Fe-N₄ sites played an important role in the further oxidization of phenol. Once the O-Fe-N₄ site was formed, a phenol was simulated to come close to it lowering total energy by van der Waals interactions. Then the C-O bond formed between phenol and O-Fe-N₄ sites, elevating energy by $\Delta E_{S3-S2}$. Finally, the desorption energy of hydroquinone (C₂H₆O₂) could be calculated as follows:

$$\Delta E_{S4-S5} = E(Fe - N_4) + E(C_2H_6O_2) - E(C_2H_6O_2 - Fe - N_4)$$

Considering the interchange of hydroquinone and benzoquinone (C₂H₄O₂) in nature, this simulated reaction path was anticipated to show key steps during catalysis.

## Data availability

The data generated in this study are provided within the article and the Supplementary Information file. All raw data generated in this study are available from the corresponding authors upon request.

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

## Acknowledgements

This work was supported by the National Postdoctoral Program for Innovative Talents of China (No. BX20220159, S.W.), the National Natural Science Foundation of China (21890383, Y.L.), (22276095, J.Q.) and Natural Science Foundation of Jiangsu Province (BK20211522, J.Q.). We thank the TPS-21A beamline for XAFS measurements in the National Synchrotron Radiation Research Center (NSRRC, Hsinchu, Taiwan).

## Author contributions

S.W., J.Q., and Y.L. conceived the idea and wrote the paper. S.W. performed the synthesis and characterization of catalysts, collected, and analyzed the data. Y.S. and J.Q. performed the Fenton-like reaction, collected and analyzed the data. Y.-Z.Q. and H.X. performed the DFT calculations. A.L. performed the in situ ETEM measurements. C.-Y.C. performed the XANES and EXAFS measurements and analyzed the data.

## Competing interests

The authors declare no competing interests.
