## [Peer Review File · Nature Communications]

REVIEWER COMMENTS

Reviewer #1 (Remarks to the Author):

In this study, Wei and coworkers described a facile strategy for carbon-defect engineering to improve the reactivity of metal ISAS catalyst in Fenton-like reaction. The authors adopted a novel self-carbon-thermal-reduction strategy to modulate the carbon-defect environment for active Fe-N₄ sites, which are well supported by solid characterizations. Moreover, the mechanism of how carbon-defect affects the PMS activation and degradation process of phenol by Fe-N₄ sites was investigated through DFT calculations. The findings of this study make important contribution to both the rational design of single-atom catalyst and the Fenton-like catalysis fields. I recommend the publication in Nat. Commun. after addressing the following issues:

1. The authors explored the removal of a variety of phenolic compounds. Could the Fe-NCv-900/PMS system degrade other organic pollutants?
2. The layout of Fig. 6 should be adjusted, for example, it is not appropriate to place Fig. 6e on top of Fig. 6d.
3. Some recently published works that contribute to the design of highly efficient Fenton-like catalysts are suggested to be cited in the introduction, such as DOI: , 10.1073/pnas.2300281120, 10.1002/anie.202303267, 10.1002/anie.202218510, and 10.1002/anie.202303728.
4. In Figure 4a, the NCv-900, Fe₂O₃ and Fe₃O₄ nano-catalysts serve as the reference samples. Therefore, the basic characterization of NCv-900, Fe₂O₃ and Fe₃O₄ nano-catalysts is necessary, such as TEM, XPS and XRD measurements.
5. The comparison of the first-order kinetic constant k in Fig. 4b, Fig. 4c and Fig. 4h should be provided.
6. After catalysis, the Fe content of Fe-NCv-900 catalyst should be provided.
7. "Phenol and phenol derivatives are important raw materials in industrial production, such as chemical production, papermaking, pharmaceutical production and coating industry, but are harmful for human being and environment." The authors should cite relative references to support above point.
8. In Supplementary Fig. 43, the author should provide the chemical structures of S0, S1, S2, S3, S4 and S5.
9. In Supplementary Fig. 10, the XRD patterns of ZIF-8 and Fe-NCv-400 should also be provided.
10. The basic characterization of ZIF-8 is necessary, such as the Zn content, TEM, XPS and XRD measurement.

Reviewer #3 (Remarks to the Author):

In this manuscript, Wei et al. presented some interesting results in synthesis of Fe single atom loaded defective carbon for catalytic activation of persulfate for organic degradation in water. In the work, the authors checked the properties of the materials and suggested the carbon defect will promote the Fe-based catalysis. However, this manuscript lacks of strong innovation in terms of materials and structure mechanism as many investigations have shown the single atom system and carbon defect in such a reaction system. In addition, some issues are not well resolved in catalysis. It is not suitable for Nat Comm. Other comments are listed below for consideration.

1. In the synthesis, the authors have tried to understand the changes of ZIF-8 for ZnO/C and Zn/C, however, they did not show any information on Fe species, which is more important. In addition, it should be presented on the role of the variation of Zn species for Fe single atom formation.
2. In all the samples, Zn metal is presented and can be active species for persulfate activation like zero valent iron.
3. For comparison, NCv-900 was tested and showed less activity. However, the structure and properties were not well characterized . Previous investigations have suggested such a material will be highly active.
4. It has been well identified that Zn metal was presented in the samples, however, no Zn-Zn bond was found in XANES characterization. Why?
5. In all the XPS spectra, no Fe was presented. The authors should use the method to identify it.
6. Similarly, Fe-NC-900 was found to be much active, while this material was either not well characterized so that it is confirmed that Fe-NC-900 did not have strong defect. The authors should check Raman spectra to get ID/IG.
7. In DFT calculation, ZnO/graphene was used, which may not be correct for a comparison. The calculation should be ZnO@Fe-N4.
8. It would be better that the authors discuss why high valent Fe was the active site and no other reaction route occurred.

General Comments:

In this paper, Li and their coworkers report a self-carbon-thermal-reduction strategy for boosting the Fenton-like activity of single Fe-N₄ sites: role of carbon-defect. Compared to other relevant reports, the present work presents a new catalyst material and an innovative catalytic mechanism. Nevertheless, there are some major concerns to be addressed before considering publication.

Specific comments:

Comment 1: On Page 8, line 10, the author mentions that the effect of carbon-defect engineering on metal ISAS catalysts in Fenton-like reactions remains unknown. Several related articles have reported on defect engineering in carbon materials and single-atom catalysts. It is important to emphasize the innovation of this article.

Comment 2: I am still unclear about the effect of Zn atoms on catalytic performance. The strong interaction between Fe single atoms and the ZnO-Carbon nano-reactor needs to be reconsidered.

Comment 3: In the abstract, the author states that carbon vacancies were easily formed adjacent to single Fe-N₄ sites during synthesis. However, the XANES and EXAFS measurements of Fe-NC_v catalysts showed more pronounced changes in the XANES spectra at the Zn K-edge. This suggests that the defective sites may be located in the second shell of Zn elements. These findings are not consistent with the results and discussion presented in the article.

Comment 4: On Page 12, line 235, the author mentions that Fe-NC_y-900 with a defective carbon substrate had more mesopores and macropores. The activity comparison should be normalized based on the specific surface area. The enhancement in catalytic performance may be attributed to the increase in surface area and the structure of the pores.

Comment 5: How to exclude the impacts of Zn element (nanoparticle、single atoms) in catalytic performance. The inert element Zinc (atomically dispersed Zn-N₄ sites) have also exhibited efficient Fenton-like chemistry.

Comment 6: Associated with question 5, more experiments and discussions should be added to define the dominant active sites, such as defective sites, single metal centers (Fe/Zn), and so on.

Comment 7: Please improve the design of the quenching experiments to scavenge the reactive species more effectively. The conclusion that high-valent iron-oxo species are the dominant reactive species needs to be supported by more rigorous evidence.

Response to the Reviewers

Reviewer #1:

Comment 1: The authors explored the removal of a variety of phenolic compounds. Could the Fe-NCv-900/PMS system degrade other organic pollutants?

Reply: We sincerely appreciate the comment provided by Reviewer #1 regarding the degradation of other organic pollutants by Fe-NCv-900/PMS system.

Therefore, we study the degradation of other organic pollutants by Fe-NCv-900/PMS system, such as aniline, Tetracycline (TC), p-Hydroxybenzoic acid (PHBA), bisphenol A (BPA), and sulfamethoxazole (SMX). As shown in **Figure R1**, the removal ratios of aniline, TC, PHBA, BPA, and SMX after degradation for 5 min are 100.0%, 73.9%, 52.1%, 48.9%, 25.3%, respectively, indicating that the high-valent iron-oxo species have different selectivity for degradation of different organic pollutants in Fe-NCv-900/PMS system, which is consistent with previous studies (*Environ. Sci. Technol.* **55**, 7034-7043 (2021); *Environ. Sci. Technol.* **54**, 14057-14065 (2020); *Environ. Sci. Technol.* **52**, 2197-2205 (2018).). We add the **Figure R1** as **Supplementary Fig. 57** in the revised **Supplementary Information**.

Figure R1. Degradation of aniline, Tetracycline (TC), p-Hydroxybenzoic acid (PHBA), bisphenol A (BPA), and sulfamethoxazole (SMX).

Comment 2: The layout of Fig. 6 should be adjusted, for example, it is not appropriate to place Fig. 6e on top of Fig. 6d.

Reply: We sincerely appreciate Reviewer #1's advice to adjust the layout of Fig. 6.

We revise the Fig. 6 as follows:

The revised Fig. 6.

Comment 3: *Some recently published works that contribute to the design of highly efficient Fenton-like catalysts are suggested to be cited in the introduction, such as DOI: 10.1073/pnas.2300281120, 10.1002/anie.202303267, 10.1002/anie.202218510, and 10.1002/anie.202303728.*

Reply: Thank you for bringing up the recently published articles on Fenton-like reactions and metal single-atom catalysis. We appreciate your suggestion to consider these references. We have carefully reviewed the articles you mentioned. These articles provide valuable insights into the field of Fenton-like reactions catalyzed by metal single-atom catalysts. We have incorporated these references into our manuscript to support and enhance our discussion on Fenton-like reactions and single-atom catalysis. We cite the references in the revised manuscript as follows: (Page 4, line 67-72.)

“Consequently, a lot of efforts have been made to develop efficient heterogeneous Fenton catalysts and use alternative oxidants especially peroxymonosulfate (PMS) to address the above issues. (10.1002/anie.202303728)”

“Various metal ISAS catalysts have exhibited great potential for PMS activation and pollutant removal (10.1073/pnas.2300281120, 10.1002/anie.202303267, and 10.1002/anie.202218510), for which optimizing the chemical environment of catalytic sites and revealing the structure-activity relationship are crucial for boosting the catalytic activity.”

Comment 4: *In Figure 4a, the NCv-900, Fe₂O₃ and Fe₃O₄ nano-catalysts serve as the reference samples. Therefore, the basic characterization of NCv-900, Fe₂O₃ and Fe₃O₄ nano-catalysts is necessary, such as TEM, XPS and XRD measurements.*

Reply: We sincerely appreciate Reviewer #1's suggestion to characterize the NCv-900, Fe₂O₃ and Fe₃O₄ nano-catalysts, such as TEM, XPS and XRD measurements.

We characterize NCv-900 catalyst by HAADF-STEM, EDX spectroscopy

elemental mapping results, XPS and XRD measurements, as shown in **Figure R2-R5**.

Figure R2. The HAADF-STEM images of NCv-900 catalyst.

Figure R3. The HAADF-STEM image and corresponding EDX spectroscopy elemental mapping results of NCv-900.

Figure R4. The XPS spectra of NCv-900. **a**, XPS spectrum for the C 1s. **b**, XPS spectrum for the N 1s. **c**, XPS spectrum for the O 1s. **d**, XPS spectrum for the Zn 2p.

For C 1s spectra, C=C, C=N and C-N bonds (*Angew. Chem. Int. Ed.* **59**, 22465-22469 (2020).) co-existed. For N 1s spectra, pyridinic N, N-metal, pyrrolic N, and graphitic N species (*Angew. Chem. Int. Ed.* **60**, 9078-9085 (2021).) co-existed. For O 1s spectra, C=O, C-O bonds (*Angew. Chem. Int. Ed.* **59**, 1961-1965 (2020).) and O-Zn bond (*Appl. Catal. B-Environ.* **310**, 121298 (2022).) co-existed.

Figure R5. The XRD spectrum of NCv-900.

Besides, the characterizations of Fe₂O₃ and Fe₃O₄ nano-catalysts by TEM, XRD and XPS measurements are shown in **Figure R6**.

Figure R6. The characterizations of Fe_2O_3 and Fe_3O_4 nano-catalysts by TEM, XRD and XPS measurements.

We add the **Figure R2-R6** as **Supplementary Figs. 34-36, 39** and **40** in the revised **Supplementary Information**.

Comment 5: The comparison of the first-order kinetic constant k in Fig. 4b, Fig. 4c and Fig. 4h should be provided.

Reply: We sincerely appreciate Reviewer #1's advice. We compare the first-order kinetic constant k in Fig. 4b, Fig. 4c and Fig. 4h in **Figure R7-R9**, respectively.

Figure R7. The comparison of the first-order kinetic constant k between 4-NP, resorcinol, 4-FP, 4-CIP, 4-BrP, 4-IP and phenol catalyzed by Fe-NCv-900.

Figure R8. The comparison of the first-order kinetic constant k for degradation of phenol between different Fe-NCv catalysts under pyrolysis at different temperatures.

Figure R9. The comparison of the first-order kinetic constant k for degradation of phenol catalyzed by Fe-NCv-900 under different pH values.

We add the **Figure R7-R9** as **Supplementary Figs. 41, 44** and **55** in the revised **Supplementary Information**.

Comment 6: After catalysis, the Fe content of Fe-NCv-900 catalyst should be provided.

Reply: We sincerely appreciate Reviewer #1's suggestion to measure the Fe content of Fe-NCv-900 catalyst after catalysis. The Fe content of Fe-NCv-900 catalyst after catalysis is 0.72 wt% by ICP-OES measurement, which is similar to the Fe content of Fe-NCv-900 catalyst (0.75 wt%). We add the Fe content of Fe-NCv-900 catalyst after catalysis in the revised **Manuscript**. (Page 17, Line 361-363)

Comment 7: "Phenol and phenol derivatives are important raw materials in industrial production, such as chemical production, papermaking, pharmaceutical production and coating industry, but are harmful for human being and environment."

The authors should cite relative references to support above point.

Reply: We sincerely appreciate Reviewer #1's suggestion to cite references relative to the degradation of phenol. We cite the references (*Environ. Sci. Technol.* **42**, 8083-8087 (2008); *Chem. Eng. J.* **385** 123732 (2020).) as **Ref. 66** and **Ref. 67** to support above point in the revised manuscript.

The corresponding description relative to the application and the toxicity of phenol from the reference (*Environ. Sci. Technol.* **42**, 8083-8087 (2008).) is shown as follows:

“Phenol is used as a general disinfectant, as a reagent in chemical analysis, and for the manufacture of artificial resins, medical and industrial organic compounds, and dyes. Phenol is also a common component of oil refinery wastes and is formed in the conversion of coal into gaseous or liquid fuels and in the production of metallurgical coke from coal. It may enter the environment by discharges from oil refineries, coal conversion plants, municipal waste treatment plants, or general spills.”

The corresponding description relative to the application and the toxicity of phenol from the reference (*Chem. Eng. J.* **385** 123732 (2020).) is shown as follows:

“Phenolic compounds are a class of pollutants with high toxicity, recalcitrance, bioaccumulation and strong odor emission, which pose a great threat to the environment and human being. Owing to their extensive use in the manufacturing process of plastics, dyes, drugs, pesticides and papers, it is urgent to develop eco-friendly methods to obliterate these contaminants in industrial wastewaters.”

Comment 8: *In Supplementary Fig. 43, the author should provide the chemical structures of S0, S1, S2, S3, S4 and S5.*

Reply: We sincerely appreciate Reviewer #1's recommendation to provide the chemical structures of S0, S1, S2, S3, S4 and S5 in **Supplementary Fig. 63**. (the **Supplementary Fig. 43** in previous version), which will make the catalytic pathways more clearly. The revised **Supplementary Fig. 63**. (the **Supplementary Fig. 43** in

previous version) is exhibited as follows:

The revised **Supplementary Fig. 63.** (the **Supplementary Fig. 43** in previous version).

Comment 9: In Supplementary Fig. 10, the XRD patterns of ZIF-8 and Fe-NCv-400 should also be provided.

Reply: We sincerely appreciate Reviewer #1's advice to provide the XRD patterns of ZIF-8 and Fe-NCv-400, which will provide additional information about carbonization of ZIF-8 and the *in-situ* formation of ZnO nanoparticles. The revised Supplementary Fig. 13 (the Supplementary Fig. 10 in previous version) is shown as follows:

The revised Supplementary Fig. 13 (the Supplementary Fig. 10 in previous version).

Comment 10: The basic characterization of ZIF-8 is necessary, such as the Zn content, TEM, XPS and XRD measurement.

Reply: We sincerely appreciate Reviewer #1's recommendation to characterize the ZIF-8 material, which is necessary to study the structural evolution of the Fe-NCv catalysts.

The Zn content of ZIF-8 is 27.20 wt% Zn, which is determined by ICP-OES measurement. The TEM and HAADF-STEM images of ZIF-8 are shown in **Figure R10** as follows:

Figure R10. a The TEM image of ZIF-8. b The HAADF-STEM image of ZIF-8.

The XPS measurement and analysis of ZIF-8 are shown in **Figure R11** as follows:

Figure R11. The XPS measurement and analysis of ZIF-8.

The XRD measurement of ZIF-8 is exhibited in **Figure R12** as follows:

Figure R12. The XRD pattern of ZIF-8.

We add the Zn content of ZIF-8 in **Supplementary Table 1**, the **Figures R10-R12** as **Supplementary Figs. 2-4** in the revised **Supplementary Information**.

Reviewer #2:

Comment 1: On Page 8, line 10, the author mentions that the effect of carbon-defect engineering on metal ISAS catalysts in Fenton-like reactions remains unknown. Several related articles have reported on defect engineering in carbon materials and single-atom catalysts. It is important to emphasize the innovation of this article.

Reply: We sincerely appreciate the reviewer's suggestion to emphasize the innovation of our work, which is important for readers to well understand the scientific significance of our work.

Therefore, we revise the second paragraph and summarize the innovation of our work in the third paragraph in the "Introduction" part as follows:

"In recent years, Fenton reaction and its derived Fenton-like reactions have received tremendous interest in various fields, including biomedicine⁴⁶, gene expression⁴⁷, biosensing⁴⁸, material and chemical synthesis^{49, 50}, and environmental remediation⁵¹. Particularly, the original version of Fenton reaction using homogeneous Fe²⁺ and H₂O₂ has achieved global application in many water treatment facilities because the generated hydroxyl radicals (\bullet OH) with strong oxidizing ability could effectively remove toxic organic pollutants⁵². However, this reaction faces several challenges including non-regenerable catalyst, narrow pH range, formation of Fe sludge, and unsafe manipulation of H₂O₂⁵³. Consequently, a lot of efforts have been made to develop efficient heterogeneous Fenton catalysts and use alternative oxidants especially peroxymonosulfate (PMS) to address the above issues^{54, 55}. Various metal ISAS catalysts have exhibited great potential for PMS activation and pollutant removal⁵⁶⁻⁶¹, for which optimizing the chemical environment of catalytic sites and revealing the structure-activity relationship are crucial for boosting the catalytic activity. Very recently, several atomic engineering strategies including oxygen vacancy⁶², oxygen doping⁶³, and coordination modulation⁶⁴ have been reported to further improve the catalytic activity of the metal ISAS catalysts. However, the effect of carbon-defect engineering of metal ISAS catalysts on Fenton-like reaction remains unexplored, which is also appealing for improving their activity. Therefore,

developing simple and low-cost synthetic methodology, rationally designing carbon-defect engineering, boosting Fenton-like activity of metal ISAS, and revealing the structure-activity relationship are meaningful for improved degradation of pollutants but with great challenges.

Herein, we design a carbon-defect engineering of single Fe-N₄ sites in ZnO-Carbon nano-reactor via self-carbon-thermal-reduction strategy, as efficient Fenton-like catalyst for degradation of phenol. By combination of experimental results and DFT calculations, we reveal that the carbon vacancies are easily constructed adjacent to single Fe-N₄ sites, facilitating the formation of C-O bonding and lowering the energy barrier of rate-determining-step during catalysis. Consequently, compared with Fe-NC-900 without abundant carbon vacancies, Fe-NCv-900 with abundant carbon vacancies exhibits a much improved activity, with 13.5 times improvement in the first-order rate constant for degradation of phenol. Besides, Fe-NCv-900 exhibits high activity (97% removal ratio of phenol in only 5 min), good recyclability and wide pH suitability (3-9). This work represents the first attempt to boost the Fenton-like activity of metal ISAS by carbon-defect engineering and to elucidate the relationship between the carbon-defect and catalytic activity in Fenton-like reaction, which are crucial for improving the degradation efficiency of organic pollutants in water remediation. The simple, cheap and robust self-carbon-thermal-reduction strategy will hopefully inspire further studies on developing novel and rational synthetic methodology in carbon-defect engineering. The combination of multiple characterization techniques, such as *in-situ* EXAFS, *in-situ* ETEM and AC-STEM measurements, provides a strong support to directly observe the structural evolution of catalysts and to reveal the structure-activity relationship during catalysis.”

Comment 2: *I am still unclear about the effect of Zn atoms on catalytic performance. The strong interaction between Fe single atoms and the ZnO-Carbon nano-reactor needs to be reconsidered.*

Reply: We sincerely appreciate the reviewer's recommendation to analyze the effect of Zn atoms on catalytic performance.

We characterize the NCv-900 catalyst without Fe loading. The NCv-900 catalyst is synthesized by co-pyrolysis of ZIF-8 and filter papers without Fe(acac)₃ at 900°C for 3h under argon atmosphere. The HAADF-STEM image and corresponding EDX spectroscopy elemental mapping results of NCv-900 are shown in **Figure R3**. The EDX spectroscopy elemental mapping results exhibit that the C, N, and Zn elements are homogeneously dispersed on the NCv-900. The Zn content of NCv-900 is 2.45 wt% Zn, which is determined by ICP-OES measurement.

Figure R3. The HAADF-STEM image and corresponding EDX spectroscopy elemental mapping results of NCv-900.

For degradation of phenol by Fenton-like reaction, Fe-NCv-900 catalyst exhibits much higher catalytic activity compared to NCv-900 catalyst without Fe loading. As shown in **Figure R13**, during degradation of phenol for 5 min, the removal ratios of phenol catalyzed by Fe-NCv-900 and NCv-900 are 97% and 10%, respectively, indicating the Fe ISAS is the catalytic site in Fe-NCv-900 catalyst during catalysis.

Figure R13. The plots of phenol concentration versus time of Fe-NCv-900 and NCv-900 catalysts.

Similarly, we also synthesize NCv-800, NCv-700 and NCv-600 without Fe loading by pyrolysis of pure ZIF-8 and filter papers without Fe(acac)₃ loading at 800°C, 700°C and 600°C, respectively. As shown in **Figure R14**, compared with Fe-NCv-800, Fe-NCv-700 and Fe-NCv-600, the catalytic activities of NCv-800, NCv-700 and NCv-600 are almost inert, which well exclude the impacts of Zn element during catalysis.

Figure R14. a, The plots of phenol concentration versus time of Fe-NCv-600 and NCv-600 catalysts. **b,** The plots of phenol concentration versus time of Fe-NCv-700 and NCv-700 catalysts. **c,** The plots of phenol concentration versus time of Fe-NCv-800 and NCv-800 catalysts.

We add the **Figure R13** and **R14** as **Supplementary Figs. 45** and **46** in the revised **Supplementary Information**.

We also sincerely appreciate the reviewer's suggestion to reconsider the strong interaction between Fe single atoms and the ZnO-Carbon nano-reactor. The interaction between metal catalytic sites and substrates plays an important role during catalysis, which has a profound effect on the catalytic activity.

The ZnO-Carbon nano-reactor serves as the reactants for self-carbon-thermal-reduction. Above 600°C, the ZnO nanoparticles are transformed into the Zn nanoparticles and Zn nanoparticles evaporate into Zn vapors, simultaneously introducing carbon-defects on the N-doped carbon substrates. Therefore, the ZnO-Carbon nano-reactor only exists below 600°C. Because the ZnO-Carbon nano-reactor does not exist in Fe-NCv-900 catalyst, the effect of carbon-defect on the catalytic activity of Fe-NCv-900 is considered during Fenton-like reactions. As demonstrated by the DFT calculations, the adjacent carbon vacancies of the Fe-N₄ site effectively facilitate the formation of C-O bonding, lower the energy barrier of rate-determining-step and boost the catalytic activity of the Fe-N₄ site during degradation of phenol.

***Comment 3:** In the abstract, the author states that carbon vacancies were easily formed adjacent to single Fe-N₄ sites during synthesis. However, the XANES and EXAFS measurements of Fe-NCv catalysts showed more pronounced changes in the XANES spectra at the Zn K-edge. This suggests that the defective sites may be located in the second shell of Zn elements. These findings are not consistent with the results and discussion presented in the article.*

Reply: We sincerely appreciate the reviewer's recommendation to consider the effect of carbon-defect on the changes of XANES spectra at the Zn K-edge.

Therefore, we construct two theoretical structures containing two different carbon vacancies in the second shell of single Zn-N₄ site on N-doped graphene in **Figure R15c**. We fit the theoretical XANES spectra of Zn-N₄/NCv-1 and Zn-N₄/NCv-2. As shown in **Figure R15a**, the spectra of Zn-N₄/NCv-1 and Zn-N₄/NCv-2 after introducing carbon vacancies in the second shell of single Zn-N₄ site have similar

peak shape with that of Zn-N₄/NC without carbon vacancies, indicating that the carbon vacancies in the second shell of single Zn-N₄ site have no profound effect on the XANES spectrum of single Zn-N₄ site. As shown in **Figure R15b**, the obvious changes of the XANES spectra from Fe-NCv samples at Zn K-edge are attributed to the structural evolution from ZnO nanoparticles to Zn single-atom sites, that is the *in-situ* formation of ZnO nanoparticles and *in-situ* disappearance of ZnO nanoparticles by self-carbon-thermal-reduction.

Figure R15. **a**, The comparison of the XANES spectra of theoretical Zn-N₄/NC, theoretical Zn-N₄/NCv-1 and theoretical Zn-N₄/NCv-2. **b**, The comparison of the XANES spectra of Fe-NCv samples at Zn K-edge, theoretical Zn-N₄/NCv-1 and theoretical Zn-N₄/NCv-2. **c**, The structural models of theoretical Zn-N₄/NCv-1 and theoretical Zn-N₄/NCv-2.

Comment 4: On Page 12, line 235, the author mentions that Fe-NCv-900 with a defective carbon substrate had more mesopores and macropores. The activity comparison should be normalized based on the specific surface area. The enhancement in catalytic performance may be attributed to the increase in surface area and the structure of the pores.

Reply: We sincerely appreciate the reviewer's advice to consider the role of surface area and the structure of the pores during catalysis.

We summarize the surface areas of Fe-NCv catalysts in **Table R1**. In order to give a fair comparison of the activities of Fe-NCv catalysts by pyrolysis at different

temperatures, we compare the activities of Fe-NCv catalysts at lower conversion of phenol (the removal ratios of phenol after degradation for 0.25 min), because the low concentration of phenol in solution at higher conversion will affect the reaction rates.

The removal ratios of phenol after degradation for 0.25 min catalyzed by different Fe-NCv catalysts are also exhibited in **Table R1**. We calculate the specific activities ($\text{mol}_{\text{phenol}}/\text{m}^2_{\text{catalyst}}$) of Fe-NC catalysts, which are normalized based on the specific surface area of catalysts as follows:

Fe-NCv-900:

$$\frac{10 \times 10^{-6} (\text{mol}/\text{L}) \times 80 \times 10^{-3} (\text{L}) \times 54.2\%}{0.8 \times 10^{-3} (\text{g}) \times 853 (\text{m}^2/\text{g})} = 6.35 \times 10^{-7} (\text{mol}_{\text{phenol}}/\text{m}^2_{\text{catalyst}})$$

Fe-NCv-800:

$$\frac{10 \times 10^{-6} (\text{mol}/\text{L}) \times 80 \times 10^{-3} (\text{L}) \times 23.2\%}{0.8 \times 10^{-3} (\text{g}) \times 573 (\text{m}^2/\text{g})} = 4.05 \times 10^{-7} (\text{mol}_{\text{phenol}}/\text{m}^2_{\text{catalyst}})$$

Fe-NCv-700:

$$\frac{10 \times 10^{-6} (\text{mol}/\text{L}) \times 80 \times 10^{-3} (\text{L}) \times 8.4\%}{0.8 \times 10^{-3} (\text{g}) \times 263 (\text{m}^2/\text{g})} = 3.19 \times 10^{-7} (\text{mol}_{\text{phenol}}/\text{m}^2_{\text{catalyst}})$$

Fe-NCv-600:

$$\frac{10 \times 10^{-6} (\text{mol}/\text{L}) \times 80 \times 10^{-3} (\text{L}) \times 3.5\%}{0.8 \times 10^{-3} (\text{g}) \times 115 (\text{m}^2/\text{g})} = 3.04 \times 10^{-7} (\text{mol}_{\text{phenol}}/\text{m}^2_{\text{catalyst}})$$

Fe-NC-900:

$$\frac{10 \times 10^{-6} (\text{mol}/\text{L}) \times 80 \times 10^{-3} (\text{L}) \times 10\%}{0.8 \times 10^{-3} (\text{g}) \times 427 (\text{m}^2/\text{g})} = 2.34 \times 10^{-7} (\text{mol}_{\text{phenol}}/\text{m}^2_{\text{catalyst}})$$

Table R1. The summary of the BET surface areas and specific activities ($\text{mol}_{\text{phenol}}/\text{m}^2_{\text{catalyst}}$) of Fe-NC catalysts.

Catalyst	BET surface area(m^2/g)	Removal ratios of phenol at 0.25 min	Specific activity ($\text{mol}_{\text{phenol}}/\text{m}^2_{\text{catalyst}}$)
Fe-NCv-600	115	3.5%	3.04×10^{-7}
Fe-NCv-700	263	8.4%	3.19×10^{-7}
Fe-NCv-800	573	23.2%	4.05×10^{-7}
Fe-NCv-900	853	54.2%	6.35×10^{-7}
Fe-NC-900	427	10.0%	2.34×10^{-7}

As shown in **Figure R16**, the specific activities ($\text{mol}_{\text{phenol}}/\text{m}^2_{\text{catalyst}}$) of Fe-NCv catalysts increase gradually from Fe-NCv-600 to Fe-NCv-900, which are higher than that of Fe-NC-900 without abundant carbon-defect. Therefore, the increasing BET surface area is not the primary factor for the increasing activities of Fe-NCv catalysts.

Figure R16. The comparison of specific activities ($\text{mol}_{\text{phenol}}/\text{m}^2_{\text{catalyst}}$) between different Fe-NC catalysts.

In order to illustrate the effect of pores on catalytic activity, we compare the distribution of pore's size and catalytic activities of Fe-NCv-700 and Fe-NC-900 catalysts. As shown in **Figure R17**, Fe-NCv-700 has more mesopores while Fe-NC-900 has more micropores (**Figure R17a** and **R17b**). Considering the

molecular diameter of phenol is around 0.69 nm and mesopores will not hinder the diffusion of phenol, therefore more micropores are advantageous for the diffusion of phenol and catalysis. Besides, the BET surface area of Fe-NC-900 is 427 m²/g, higher than that of Fe-NCv-700 with the BET surface area of 263 m²/g. The Fe loading of Fe-NC-900 is 0.78 wt% Fe, also higher than that of Fe-NCv-700 with 0.56 wt% Fe. Therefore, compared with Fe-NCv-700, the more micropores, larger BET surface area and higher Fe loading of Fe-NC-900 are more advantageous for boosting catalytic activity. However, for degradation of phenol by Fenton-like reaction, Fe-NCv-700 exhibits higher activity than that of Fe-NC-900, as shown in **Figure R17c**, indicating that the structure of pores and the BET surface area are not the major factor during catalysis.

Figure R17. The comparison of distribution of **a** mesopores, **b** micropores and **c** catalytic activity between Fe-NCv-700 and Fe-NC-900.

We add the **Figure R16** and **R17** as **Supplementary Figs. 47** and **48** in the revised **Supplementary Information**.

Comment 5: How to exclude the impacts of Zn element (nanoparticle, single atoms) in catalytic performance. The inert element Zinc (atomically dispersed Zn-N₄ sites) have also exhibited efficient Fenton-like chemistry.

Reply: We sincerely appreciate the reviewer's comment to exclude the impacts of Zn element in catalysis. We also synthesize and characterize the NCv-900 catalyst without Fe loading. The NCv-900 catalyst is synthesized by co-pyrolysis of ZIF-8 and

filter papers without $\text{Fe}(\text{acac})_3$ at 900°C for 3h under argon atmosphere. The Zn content of NCv-900 is 2.45 wt% Zn, measured by ICP-OES measurement. As shown in **Figure R3**, the HAADF-STEM image and the corresponding EDX spectroscopy elemental mapping results exhibit the homogeneous distribution of C, N, and Zn element over the N-doped carbon substrate.

Figure R3. The HAADF-STEM image and corresponding EDX spectroscopy elemental mapping results of NCv-900.

During Fenton-like reaction for degradation of phenol, Fe-NCv-900 catalyst has much higher catalytic activity compared to NCv-900 catalyst without Fe loading. As exhibited in **Figure R13**, after 5 min for degradation of phenol, the removal ratios of phenol catalyzed by Fe-NCv-900 and NCv-900 are 97% and 10%, respectively, demonstrating that the Fe ISAS is the catalytic site in Fe-NCv-900 catalyst during catalysis.

Figure R13. The plots of phenol concentration versus time of Fe-NCv-900 and NCv-900 catalysts.

Similarly, we also synthesize NCv-800, NCv-700 and NCv-600 without Fe loading by pyrolysis of pure ZIF-8 and filter papers without Fe(acac)₃ loading at 800°C, 700°C and 600°C, respectively. As shown in **Figure R14**, compared with Fe-NCv-800, Fe-NCv-700 and Fe-NCv-600, the catalytic activities of NCv-800, NCv-700 and NCv-600 are almost inert, which well exclude the impacts of Zn element during catalysis.

Figure R14. **a**, The plots of phenol concentration versus time of Fe-NCv-600 and NCv-600 catalysts. **b**, The plots of phenol concentration versus time of Fe-NCv-700 and NCv-700 catalysts. **c**, The plots of phenol concentration versus time of Fe-NCv-800 and NCv-800 catalysts.

Recently, other works also reported that the N-doped carbons by pyrolysis of pure ZIF-8 exhibited poor catalytic activities for Fenton-like reactions.

For instance, Tian et al. (*Environmental Functional Materials* **1**, 267-274 (2022).)

reported the microporous nitrogen-doped nanocarbons from the carbonization of ZIF-8 (ZCN) effectively boosted the degradation efficiency of sulfamethoxazole (SMX) in Fe(III)/H₂O₂ system. However, the catalytic activity of ZCN/H₂O₂ system was rather poor.

As shown in **Figure R18** (the Fig. 2a in *Environmental Functional Materials* **1**, 267-274 (2022).), “The removal efficiency of SMX shown in Fig. 2a via ZCN adsorption was less than 18%. The unimproved SMX degradation efficiency by both ZCN/Fe(III) (20%) and ZCN/H₂O₂ (21%) reflected that ZCN did not own any capability to active Fe(III) or H₂O₂ alone.” indicate that the ZCN obtained by pyrolysis of ZIF-8 can not activate H₂O₂ for degradation of SMX.

Figure R18. The Fig. 2a in *Environmental Functional Materials* **1**, 267-274 (2022).

In 2022, Zhu et al. (*Journal of Environmental Chemical Engineering* **10**, 107758 (2022).) reported ZIF-8-derived single-atom Cu and N co-coordinated porous carbon as bifunctional material for SMX removal. As shown in **Figure R19** (the Fig. 6 in *Journal of Environmental Chemical Engineering* **10**, 107758 (2022).), when N-C (by pyrolysis of ZIF-8) and Cu-N-C are added separately, the adsorption removal rate of SMX are 66.1% and 65.2%, respectively. When adding PMS during catalysis, the total SMX degradation rates of N-C/PMS and Cu-N-C/PMS are 75.8% and 100%, respectively, indicating that the removal rates of SMX degradation catalyzed by N-C/PMS and Cu-N-C/PMS are 9.7% and 34.8%, respectively, demonstrating the N-C catalyst obtained by pyrolysis of ZIF-8 exhibits rather poor activity for

Fenton-like reactions.

Figure R19. The Fig. 6 in *Journal of Environmental Chemical Engineering* **10**, 107758 (2022).

Besides, other works also reported the poor catalytic activities of ZIF-8-derived CN materials for Fenton-like reactions, such as the Fig. 5a from the reference (*Chem. Eng. J.* **451**, 138597 (2023).) and the Fig. 6a from the reference (*Chem. Eng. J.* **419**, 129590 (2021).).

Comment 6: Associated with question 5, more experiments and discussions should be added to define the dominant active sites, such as defective sites, single metal centers (Fe/Zn), and so on.

Reply: We sincerely appreciate the reviewer's recommendation to define the dominant catalytic active sites.

We synthesize NCv-900 catalyst without Fe loading as the reference sample. The NCv-900 catalyst is synthesized by co-pyrolysis of ZIF-8 and filter papers without Fe(acac)₃ at 900°C for 3h under argon atmosphere. We characterize the NCv-900 catalyst in detail as follows:

Figure R2. The HAADF-STEM images of NCv-900 catalyst.

The HAADF-STEM images of NCv-900 catalyst are shown in **Figure R2**. As shown in **Figure R20**, the BET surface area of NCv-900 is 648 m²/g. The NCv-900 catalyst also has abundant mesopores and micropores, the same as Fe-NCv-900 catalyst, measured by nitrogen sorption isotherm experiments, as shown in **Figure R20**. We add the **Figure R20** as **Supplementary Fig. 37** in the revised **Supplementary Information**.

Figure R20. The BET surface area and pore size distribution of NCv-900. **a**, N₂ adsorption-desorption isotherms and corresponding BET surface area. **b**, The comparison of mesopore-size distribution of NCv-900 and Fe-NCv-900. **c**, The comparison of micropore-size distribution of NCv-900 and Fe-NCv-900.

Figure R3. The HAADF-STEM image and corresponding EDX spectroscopy elemental mapping results of NCv-900.

The EDX spectroscopy elemental mapping results of NCv-900 in **Figure R3** demonstrate the existence of Zn element and the Zn content of NCv-900 is 2.45 wt% Zn, determined by ICP-OES measurement.

Figure R21. **a**, The Raman spectrum of NCv-900. **b**, The EPR spectrum of NCv-900.

The Raman spectrum of NCv-900 is shown in **Figure R21a**. The I_D/I_G value of NCv-900 catalyst is 1.04, similar to those of Fe-NCv-800 ($I_D/I_G = 1.05$) and Fe-NCv-900 ($I_D/I_G = 1.07$), indicating the existence of disorder structure of NCv-900. The EPR spectrum of NCv-900 is shown in **Figure R21b**, with an obvious signal at around 2.005 g, indicating that the existence of carbon-vacancies in the carbon substrate of NCv-900. (*Adv. Mater.* **35**, 2210714 (2023).) We add the **Figure R21** as **Supplementary Fig. 38** in the revised **Supplementary Information**.

Therefore, except for absence of Fe element, NCv-900 catalyst has the same

characteristic with the Fe-NCv-900 catalyst, with the existence of Zn element, abundant mesopores, disorder structure and carbon-vacancies in the carbon substrate.

Figure R13. The plots of phenol concentration versus time of Fe-NCv-900 and NCv-900 catalysts.

For degradation of phenol by Fenton-like reaction, Fe-NCv-900 catalyst exhibits much higher catalytic activity compared to NCv-900 catalyst without Fe loading. As shown in **Figure R13**, during degradation of phenol for 5 min, the removal ratios of phenol catalyzed by Fe-NCv-900 and NCv-900 are 97% and 10%, respectively, indicating that the Fe ISAS is the dominant catalytic site rather than the Zn species or the carbon-vacancies in Fe-NCv-900 catalyst during catalysis.

***Comment 7:** Please improve the design of the quenching experiments to scavenge the reactive species more effectively. The conclusion that high-valent iron-oxo species are the dominant reactive species needs to be supported by more rigorous evidence.*

Reply: We sincerely appreciate the reviewer's advice to improve the design of the quenching experiments to confirm the high-valent iron-oxo species are the dominant reactive species.

Therefore, additional inhibition experiment is performed to confirm the high-valent iron-oxo species as active sites. Firstly, we exclude the radical and singlet oxygen (¹O₂) as ROS by quenching experiments in **Supplementary Figs. 49-51**. We add 100 mM Dimethyl sulfoxide (DMSO) as the inhibitor of high-valent iron-oxo

species in Fe-NCv-900/PMS system for degradation of phenol. As reported by the reference (*Angew. Chem.* **117**, 7031-7034 (2005).), the DMSO can consume the high-valent iron-oxo species by oxygen-atom-transfer step.

As shown in **Figure R22**, after adding 100 mM Dimethyl sulfoxide (DMSO) as the inhibitor of high-valent iron-oxo species, the removal ratio of phenol is 42.0% after 5 min for degradation, with an obvious decline compared to the control group without DMSO (97% removal ratio of phenol after 5 min for degradation), indicating that the decreasing activity of Fe-NCv-900 is attributed to the inhibition of DMSO for high-valent iron-oxo species, which is consistent with previous studies (*Appl. Catal. B: Environ.* **305**, 123049 (2022); *Chem. Eng. J.* **427**, 130803 (2022).). We add the **Figure R22** as **Supplementary Fig. 52** in the revised **Supplementary Information**.

Figure R22. The inhibition experiment of the high-valent iron-oxo species by adding 100 mM Dimethyl sulfoxide (DMSO) as the inhibitor of high-valent iron-oxo species in Fe-NCv-900/PMS system for degradation of phenol.

Besides, we exclude the radical and singlet oxygen ($^1\text{O}_2$) as ROS by quenching experiments as follows:

“We further investigated the reactive species for Fenton-like reactions. In **Supplementary Fig. 49**, we explored the reactive oxygen species (ROS) in the Fe-NCv-900/PMS system by EPR experiments with 5,5-dimethyl-1-pyrroline N-oxide (DMPO) as a trapping agent. No signals appeared when PMS or Fe-NCv-900 solely

existed while Fe-NCv-900/PMS system exhibited a strong characteristic signal of 5,5-dimethyl-1-pyrrolidone-2-oxyl (DMPOX) with peak intensity of 1:2:1:2:1:2:1 (*Adv. Mater.* **34**, 2110653 (2022).). Previous study suggested that DMPO could be oxidized into DMPOX by radical or non-radical pathway (*Adv. Mater.* **34**, 2110653 (2022); *Angew. Chem. Int. Ed.* **61**, e202200406 (2022); *Environ. Sci. Technol.* **54**, 3714-3724 (2020); *Appl. Catal. B: Environ.* **241**, 561-569 (2019).).

Supplementary Fig. 49. The EPR experiments with 5,5-dimethyl-1-pyrroline N-oxide (DMPO) as a trapping agent. **a**, Fe-NCv-900/PMS system ($[\text{catalyst}]_0 = 10$ mg/L, $[\text{DMPO}]_0 = 100$ mM, $[\text{PMS}]_0 = 100$ μM , initial pH = 7.0 ± 0.1). **b**, only Fe-NCv-900 ($[\text{catalyst}]_0 = 10$ mg/L, $[\text{DMPO}]_0 = 100$ mM, initial pH = 7.0 ± 0.1). **c**, only PMS ($[\text{DMPO}]_0 = 100$ mM, $[\text{PMS}]_0 = 100$ μM , initial pH = 7.0 ± 0.1).

Then we added 100 mM tert-butanol (TBA) or ethanol (EtOH) as quenching agents to scavenge radicals species (*Adv. Mater.* **34**, 2110653 (2022).) into Fe-NCv-900/PMS system. The degradation of phenol was not inhibited after adding TBA or EtOH, demonstrating radical was not ROS in this system (**Supplementary Fig. 50**).

Supplementary Fig. 50. The quenching experiments with TBA or EtOH as quenching agents to scavenge HO[•] and SO₄^{•-} radicals. ([catalyst]₀ = 10 mg/L, [Phenol]₀ = 10 μM, [PMS]₀ = 100 μM, initial pH = 7.0±0.1)

Except for radical as ROS, singlet oxygen (¹O₂) (*Angew. Chem. Int. Ed.* **61**, e202202338 (2022).) and high-valent metals species (*Adv. Mater.* **34**, 2110653 (2022); *Proc. Natl. Acad. Sci. USA.* **120**, e2219923120 (2023).) as ROS were reported for activation of PMS. When using 2,2,6,6-tetramethylpiperidine (TEMP) as spin-trapping agent for ¹O₂ (*Adv. Mater.* **34**, 2110653 (2022); *Angew. Chem. Int. Ed.* **61**, e202202338 (2022).), no signals were detected in **Supplementary Fig. 51**, excluding ¹O₂ as ROS.

Supplementary Fig. 51. The EPR spectrum with TEMP as spin trapping agents for singlet oxygen (¹O₂). ([catalyst]₀ = 10 mg/L, [TEMP]₀ = 100 mM, [PMS]₀ = 100 μM, initial pH = 7.0 ± 0.1)

In Fig. 4g, methyl phenyl sulfoxide (PMSO) as a chemical probe of high-valent iron-oxo species (*Adv. Mater.* **34**, 2110653 (2022).) was added in Fe-NCv-900/PMS system. The peak of methyl phenyl sulfone (PMSO₂) by high performance liquid chromatography (HPLC) measurement demonstrated the formation of high-valent iron-oxo species for activation of PMS. Therefore, we concluded that high-valent iron-oxo species was the ROS in Fe-NCv-900/PMS system, because PMSO was oxidized into PMSO₂ by high-valent iron-oxo species.”

Fig. 4g High performance liquid chromatography (HPLC) spectra with PMSO as a chemical probe of high-valent iron-oxo species.

We sincerely appreciate the reviewer’s suggestion to provide more rigorous evidence of high-valent Fe as catalytic sites. Therefore, we perform the *in-situ* XANES and EXAFS measurements at Fe K-edge and revise the Manuscript as follows:

“To test the existence form of Fe catalytic sites from Fe-NCv-900 during Fenton-like reaction, *in-situ* XANES and EXAFS measurements were performed. The Fe-NCv-900 catalyst on carbon paper was immersed into 10 μ M phenol and 100 μ M PMS aqueous solution during *in-situ* Fenton-like reaction. In Fig. 5a, the XANES spectra of Fe-NCv-900 and Fe-NCv-900 during *in-situ* Fenton-like reaction exhibited

similar characteristic with that of FePc, with the pre-edge peak around 7114 eV assigned to the centrosymmetric Fe-N₄ square-planar structure (*Nat. Commun.* **11**, 4173 (2020).), indicating the Fe element existed as Fe ISAS during catalysis. As shown in the insert of **Fig. 5b**, the energy of the adsorption edge of Fe-NCv-900 during *in-situ* Fenton-like reaction exhibited an obvious positive shift compared to Fe-NCv-900. Besides, the white line intensity around 7135 eV of Fe-NCv-900 during *in-situ* Fenton-like reaction was higher than that of Fe-NCv-900, confirming the formation of high-valent iron-oxo species. (*Adv. Mater.* **34**, 2110653 (2022).) The corresponding Fourier-transform EXAFS (FT-EXAFS) spectra in R space were shown in **Fig. 5c**. During *in-situ* Fenton-like reaction, the intensity of the dominant peak of Fe-NCv-900 at around 1.5 Å obviously increased compared to Fe-NCv-900 catalyst, which was assigned to the *in-situ* formation of Fe-O bond from high-valent iron-oxo species. Compared with the WT contour plot of Fe-NCv-900 in **Fig. 5d**, the intensity of the dominant peak (4.5 Å⁻¹ in k space, 1.5 Å in R space) of Fe-NCv-900 during *in-situ* Fenton-like reaction in **Fig. 5e** obviously increased and the spatial distribution of the dominant peak during *in-situ* Fenton-like reaction moved towards higher k space, owing to the formation of Fe-O bonds from high-valent iron-oxo species during catalysis, which was similar to the Fe-O pathway from Fe₂O₃ in **Fig. 5f**. Compared with Fe₂O₃, the absence of Fe-O-Fe pathway (8.5 Å⁻¹, 3.0 Å) from Fe-NCv-900 during *in-situ* Fenton-like reaction demonstrated the sole existence of Fe-ISAS, revealing the excellent stability of Fe-ISAS from Fe-NCv-900 during catalysis.”

Fig. 5 The *in-situ* XANES and EXAFS measurements for Fenton-like reaction. **a** The XANES spectra of FePc, Fe-NCv-900, and Fe-NCv-900 during *in-situ* Fenton-like reaction. **b** The comparison of XANES spectra of Fe-NCv-900 and Fe-NCv-900 during *in-situ* Fenton-like reaction. **c** The corresponding FT-EXAFS spectra in R space of Fe-NCv-900 and Fe-NCv-900 during *in-situ* Fenton-like reaction. **d-f** The WT analysis of Fe-NCv-900, Fe-NCv-900 during *in-situ* Fenton-like reaction and Fe₂O₃.

Therefore, the high-valent iron-oxo species are demonstrated as the active sites during Fenton-like reaction for degradation of phenol by combination of *in-situ* XANES and EXAFS measurements, quenching experiments which exclude the radical and singlet oxygen ($^1\text{O}_2$) as ROS, and inhibition experiment by DMSO.

Reviewer #3:

Comment: In this manuscript, Wei et al. presented some interesting results in synthesis of Fe single atom loaded defective carbon for catalytic activation of persulfate for organic degradation in water. In the work, the authors checked the properties of the materials and suggested the carbon defect will promote the Fe-based catalysis. However, this manuscript lacks of strong innovation in terms of materials and structure mechanism as many investigations have shown the single atom system and carbon defect in such a reaction system. In addition, some issues are not well resolved in catalysis. It is not suitable for Nat Comm. Other comments are listed below for consideration.

Reply: We sincerely appreciate the reviewer's comment to improve the description of the innovation of our work.

Therefore, we revise the second paragraph and summarize the innovation of our work in the third paragraph in the "Introduction" part as follows:

"In recent years, Fenton reaction and its derived Fenton-like reactions have received tremendous interest in various fields, including biomedicine⁴⁶, gene expression⁴⁷, biosensing⁴⁸, material and chemical synthesis^{49, 50}, and environmental remediation⁵¹. Particularly, the original version of Fenton reaction using homogeneous Fe^{2+} and H_2O_2 has achieved global application in many water treatment facilities because the generated hydroxyl radicals ($\bullet\text{OH}$) with strong oxidizing ability could effectively remove toxic organic pollutants⁵². However, this reaction faces several challenges including non-regenerable catalyst, narrow pH range, formation of Fe sludge, and unsafe manipulation of H_2O_2 ⁵³. Consequently, a lot of efforts have been made to develop efficient heterogeneous Fenton catalysts and use alternative oxidants especially peroxymonosulfate (PMS) to address the above issues^{54, 55}. Various metal ISAS catalysts have exhibited great potential for PMS activation and pollutant removal⁵⁶⁻⁶¹, for which optimizing the chemical environment of catalytic sites and revealing the structure-activity relationship are crucial for boosting the catalytic activity. Very recently, several atomic engineering strategies including oxygen

vacancy⁶², oxygen doping⁶³, and coordination modulation⁶⁴ have been reported to further improve the catalytic activity of the metal ISAS catalysts. However, the effect of carbon-defect engineering of metal ISAS catalysts on Fenton-like reaction remains unexplored, which is also appealing for improving their activity. Therefore, developing simple and low-cost synthetic methodology, rationally designing carbon-defect engineering, boosting Fenton-like activity of metal ISAS, and revealing the structure-activity relationship are meaningful for improved degradation of pollutants but with great challenges.

Herein, we design a carbon-defect engineering of single Fe-N₄ sites in ZnO-Carbon nano-reactor via self-carbon-thermal-reduction strategy, as efficient Fenton-like catalyst for degradation of phenol. By combination of experimental results and DFT calculations, we reveal that the carbon vacancies are easily constructed adjacent to single Fe-N₄ sites, facilitating the formation of C-O bonding and lowering the energy barrier of rate-determining-step during catalysis. Consequently, compared with Fe-NC-900 without abundant carbon vacancies, Fe-NCv-900 with abundant carbon vacancies exhibits a much improved activity, with 13.5 times improvement in the first-order rate constant for degradation of phenol. Besides, Fe-NCv-900 exhibits high activity (97% removal ratio of phenol in only 5 min), good recyclability and wide pH suitability (3-9). This work represents the first attempt to boost the Fenton-like activity of metal ISAS by carbon-defect engineering and to elucidate the relationship between the carbon-defect and catalytic activity in Fenton-like reaction, which are crucial for improving the degradation efficiency of organic pollutants in water remediation. The simple, cheap and robust self-carbon-thermal-reduction strategy will hopefully inspire further studies on developing novel and rational synthetic methodology in carbon-defect engineering. The combination of multiple characterization techniques, such as *in-situ* EXAFS, *in-situ* ETEM and AC-STEM measurements, provides a strong support to directly observe the structural evolution of catalysts and to reveal the structure-activity relationship during catalysis.”

Comment 1: In the synthesis, the authors have tried to understand the changes of ZIF-8 for ZnO/C and Zn/C, however, they did not show any information on Fe species, which is more important. In addition, it should be presented on the role of the variation of Zn species for Fe single atom formation.

Reply: We sincerely appreciate the reviewer’s suggestion to provide additional information on Fe species. Therefore, we perform the wavelet transform (WT) analysis at Fe K-edge due to its powerful resolutions in both k and R space. As shown in **Figure R23**, we compare the contour plots of Fe-NCv catalysts, Fe₂O₃ and Fe foil as reference samples. In **Figure R23g**, the dominant peak at around 8.0 Å⁻¹ in k space and 2.2 Å in R space of Fe foil is ascribed to Fe-Fe bond. In **Figure R23h**, the dominant peak at around 8.5 Å⁻¹ in k space and 3.0 Å in R space of Fe₂O₃ is ascribed to Fe-O-Fe bond while the secondary dominant peak at around 5.0 Å⁻¹ in k space and 1.5 Å in R space of Fe₂O₃ is ascribed to Fe-O bond. By comparison, all the Fe-NCv catalysts only have one prominent peak at around 5.0 Å⁻¹ in k space and 1.5 Å in R space assigning to Fe-N bond, without the characteristic peaks of Fe-Fe bond or Fe-O-Fe bond, indicating that the Fe element of the Fe-NCv catalysts exists as isolated single-atom sites. We add the **Figure R23** as **Supplementary Fig. 29** and above analysis in the revised **Supplementary Information**.

Figure R23. The WT analysis at Fe K-edge of Fe-NCv catalysts, Fe foil and Fe₂O₃ as reference samples.

We sincerely appreciate the reviewer's comment to discuss the role of the variation of Zn species for Fe single atom formation. During the *in-situ* formation of ZnO nanoparticles, the *in-situ* released water vapor from filter papers reacts with Zn²⁺ ions from ZIF-8, with the breaking of Zn-N bonds and agglomeration of Zn element. The coordination-unsaturated N atoms from the broken Zn-N bonds serve as the anchoring sites of Fe species, with the formation of Fe-ISAS. During the *in-situ* disappearance of ZnO nanoparticles by self-carbon-thermal-reduction in ZnO-Carbon nano-reactor, the carbon vacancies are introduced around Fe-ISAS, which effectively boost the catalytic activity of Fe-ISAS for Fenton-like reactions. Therefore, the *in-situ* formation of ZnO nanoparticles facilitates the formation of Fe-ISAS and the *in-situ* disappearance of ZnO nanoparticles induces the formation of carbon vacancies around Fe ISAS.

We add the above analysis in **Supplementary Fig. 29** of the revised **Supplementary Information**.

Comment 2: In all the samples, Zn metal is presented and can be active species for persulfate activation like zero valent iron.

Reply: We sincerely appreciate the reviewer's advice to discuss whether the Zn element serves as the active species during catalysis.

We characterize the NCv-900 catalyst without Fe loading. The NCv-900 catalyst is synthesized by co-pyrolysis of ZIF-8 and filter papers without Fe(acac)₃ at 900°C for 3h under argon atmosphere. The HAADF-STEM image and corresponding EDX spectroscopy elemental mapping results of NCv-900 are shown in **Figure R3**. The EDX spectroscopy elemental mapping results exhibit that the C, N, and Zn elements are homogeneously dispersed on the NCv-900. The Zn content of NCv-900 is 2.45 wt% Zn, which is determined by ICP-OES measurement.

Figure R3. The HAADF-STEM image and corresponding EDX spectroscopy elemental mapping results of NCv-900.

For degradation of phenol by Fenton-like reaction, Fe-NCv-900 catalyst exhibits much higher catalytic activity compared to NCv-900 catalyst without Fe loading. As shown in **Figure R13**, during degradation of phenol for 5 min, the removal ratios of phenol catalyzed by Fe-NCv-900 and NCv-900 are 97% and 10%, respectively, indicating the Fe ISAS is the catalytic site in Fe-NCv-900 catalyst during catalysis.

Figure R13. The plots of phenol concentration versus time of Fe-NCv-900 and NCv-900 catalysts.

Similarly, we also synthesize NCv-800, NCv-700 and NCv-600 without Fe loading by pyrolysis of pure ZIF-8 and filter papers without Fe(acac)₃ loading at 800°C, 700°C and 600°C, respectively. As shown in **Figure R14**, compared with Fe-NCv-800, Fe-NCv-700 and Fe-NCv-600, the catalytic activities of NCv-800, NCv-700 and NCv-600 are almost inert, which well exclude the impacts of Zn element during catalysis.

Figure R14. **a**, The plots of phenol concentration versus time of Fe-NCv-600 and NCv-600 catalysts. **b**, The plots of phenol concentration versus time of Fe-NCv-700 and NCv-700 catalysts. **c**, The plots of phenol concentration versus time of Fe-NCv-800 and NCv-800 catalysts.

Recently, other works also report that the N-doped carbons by pyrolysis of pure ZIF-8 exhibit poor catalytic activities for Fenton-like reactions.

For instance, Tian et al. (*Environmental Functional Materials* **1**, 267-274 (2022).) reported the microporous nitrogen-doped nanocarbons from the carbonization of ZIF-8 (ZCN) effectively boosted the degradation efficiency of sulfamethoxazole (SMX) in Fe(III)/H₂O₂ system. However, the catalytic activity of ZCN/H₂O₂ system is rather poor.

As shown in **Figure R18** (the Fig. 2a in *Environmental Functional Materials* **1**, 267-274 (2022).), “The removal efficiency of SMX shown in Fig. 2a via ZCN adsorption was less than 18%. The unimproved SMX degradation efficiency by both ZCN/Fe(III) (20%) and ZCN/H₂O₂ (21%) reflected that ZCN did not own any capability to active Fe(III) or H₂O₂ alone.” indicate that the ZCN obtained by pyrolysis of ZIF-8 can not activate H₂O₂ for degradation of SMX.

Figure R18. The Fig. 2a in *Environmental Functional Materials* **1**, 267-274 (2022).

In 2022, Zhu et al. (*Journal of Environmental Chemical Engineering* **10**, 107758 (2022).) reported ZIF-8-derived single-atom Cu and N co-coordinated porous carbon as bifunctional material for SMX removal. As shown in **Figure R19** (the Fig. 6 in *Journal of Environmental Chemical Engineering* **10**, 107758 (2022).), when N-C (by pyrolysis of ZIF-8) and Cu-N-C are added separately, the adsorption removal rate of SMX are 66.1% and 65.2%, respectively. When adding PMS during catalysis, the total SMX degradation rates of N-C/PMS and Cu-N-C/PMS are 75.8% and 100%, respectively, indicating that the removal rates of SMX degradation catalyzed by N-C/PMS and Cu-N-C/PMS are 9.7% and 34.8%, respectively, demonstrating the N-C catalyst obtained by pyrolysis of ZIF-8 exhibits rather poor activity for Fenton-like reactions.

Figure R19. The Fig. 6 in *Journal of Environmental Chemical Engineering* **10**, 107758 (2022).

Besides, other works also reported the poor catalytic activities of ZIF-8-derived CN materials for Fenton-like reactions, such as the Fig. 5a from the reference (*Chem. Eng. J.* **451**, 138597 (2023).) and the Fig. 6a from the reference (*Chem. Eng. J.* **419**, 129590 (2021).).

Comment 3: For comparison, NCv-900 was tested and showed less activity. However, the structure and properties were not well characterized. Previous investigations have suggested such a material will be highly active.

Reply: We sincerely appreciate the reviewer's suggestion to characterize the NCv-900 catalyst in detail. The NCv-900 catalyst is synthesized by co-pyrolysis of ZIF-8 and filter papers without $\text{Fe}(\text{acac})_3$ at 900°C for 3h under argon atmosphere. We characterize the NCv-900 catalyst in detail as follows:

Figure R2. The HAADF-STEM images of NCv-900 catalyst.

The HAADF-STEM images of NCv-900 catalyst are shown in **Figure R2**. As shown in **Figure R20**, the BET surface area of NCv-900 is $648 \text{ m}^2/\text{g}$. The NCv-900 catalyst also has abundant mesopores and micropores, the same as Fe-NCv-900 catalyst, measured by nitrogen sorption isotherm experiments, as shown in **Figure R20**.

Figure R20. The BET surface area and pore size distribution of NCv-900. **a**, N_2 adsorption-desorption isotherms and corresponding BET surface area. **b**, The comparison of mesopore-size distribution of NCv-900 and Fe-NCv-900. **c**, The comparison of micropore-size distribution of NCv-900 and Fe-NCv-900.

Figure R3. The HAADF-STEM image and corresponding EDX spectroscopy elemental mapping results of NCv-900.

The EDX spectroscopy elemental mapping results of NCv-900 in **Figure R3** demonstrate the existence of Zn element and the Zn content of NCv-900 is 2.45 wt% Zn, determined by ICP-OES measurement.

Figure R21. a, The Raman spectrum of NCv-900. b, The EPR spectrum of NCv-900.

The Raman spectrum of NCv-900 is shown in **Figure R21a**. The I_D/I_G value of NCv-900 catalyst is 1.04, similar to those of Fe-NCv-800 (I_D/I_G = 1.05) and Fe-NCv-900 (I_D/I_G = 1.07), indicating the existence of disorder structure of NCv-900. The EPR spectrum of NCv-900 is shown in **Figure R21b**, with an obvious signal at around 2.005 g, indicating that the existence of carbon-vacancies in the carbon substrate of NCv-900. (*Adv. Mater.* **35**, 2210714 (2023).)

Therefore, except for absence of Fe element, NCv-900 catalyst has similar characteristic with the Fe-NCv-900 catalyst, with the existence of Zn element, abundant mesopores, disorder structure and carbon-vacancies in the carbon substrate.

Figure R13. The plots of phenol concentration versus time of Fe-NCv-900 and NCv-900 catalysts.

For degradation of phenol by Fenton-like reaction, Fe-NCv-900 catalyst exhibits much higher catalytic activity compared to NCv-900 catalyst without Fe loading. As

shown in **Figure R13**, during degradation of phenol for 5 min, the removal ratios of phenol catalyzed by Fe-NCv-900 and NCv-900 are 97% and 10%, respectively, indicating the Fe ISAS is the catalytic site rather than the Zn species or the carbon-vacancies in Fe-NCv-900 catalyst during catalysis.

The synthetic methods, size-distribution and morphology of N-doped carbon-based materials will have a profound effect on their catalytic activity for Fenton-like reactions. Recently, other works also report that the N-doped carbon-based materials as the substrates exhibit poor catalytic activities for Fenton-like reactions.

For example, Xiong et al. (*Adv. Mater.* **34**, 2110653 (2022).) reported that single-Atom Fe on CN substrate had excellent activity for degradation of bisphenol A (BPA) by Fenton-like reaction. By contrast, as shown in **Figure R24**, “The catalytic properties of the nitrogen-doped carbon support (synthesized using the same strategy) and N-graphene are both inert to this reaction.”

Figure R24. The Figure 4a from *Adv. Mater.* **34**, 2110653 (2022).

In 2021, Su et al. (*Angew. Chem. Int. Ed.* **60**, 21261-21266 (2021).) reported the Fe single-atom sites on CN substrate as efficient catalyst for degradation of ciprofloxacin (CIP) by Fenton-like reaction. By comparison, the CN substrates exhibited poor Fenton-like activities, as shown in **Figure R25**. “The CIP removal of CN and Nv/CN reach 15% and 27% within 60 min, respectively. Additionally, Fe₁/CN

shows a great improvement on CIP degradation, mainly because Fe-N₄ center can behave as active sites for H₂O₂ activation.”

Figure R25. The Figure 3b from *Angew. Chem. Int. Ed.* **60**, 21261-21266 (2021).

In 2022, Zhang et al. (*Appl. Catal. B: Environ.* **315**, 121536 (2022).) reported an advanced single-atomic iron-loaded graphitic carbon nitride (Fe₁/CN) with outstanding Fenton-like activity for degradation of Atrazine (ATZ). By contrast, the CN substrates exhibited poor catalytic activity for Fenton-like reaction. As shown in **Figure R26**, “For the CN and acid-washed CN (CNW) catalysts, only about 10% of ATZ was degraded within 60 min, indicating their weak PMS activation ability”.

Figure R26. The Fig. 1a from *Appl. Catal. B: Environ.* **315**, 121536 (2022).

In 2023, Zhang et al. (*Chem. Eng. J.* **451**, 138574 (2023).) reported a nitrogenized graphitic carbon matrix containing Fe-Co dual single atoms (FeCoNC) with excellent Fenton-like activity for degradation of tetracycline hydrochloride (TC). While the NC

substrate had poor catalytic activity for degradation of TC. As shown in **Figure R27**, “Without addition of single Fe/Co atom catalysts, individual PMS and N–C displayed inappreciable TC degradation efficiency.”

Figure R27. The Fig. 3a from *Chem. Eng. J.* **451**, 138574 (2023).

Comment 4: It has been well identified that Zn metal was presented in the samples, however, no Zn-Zn bond was found in XANES characterization. Why?

Reply: We sincerely appreciate the reviewer’s comment. We will discuss the formation of Zn nanoparticles derived from the ZnO nanoarticles in detail.

During the self-carbon-thermal-reduction in ZnO-Carbon nano-reactor, ZnO nanoparticles serve as nano-oxidant and carbon nano-cages serve as nano-reductant. Therefore, the Zn nanoparticles will be formed by reduction of carbon element. Because the boiling point of bulk zinc is 907°C and the Zn nanoparticles are much easier to evaporate than bulk zinc, the Zn nanoparticles as the intermediate product are unstable.

Therefore, we only observe the Zn nanoparticles as the unstable intermediate product just **for a few seconds** after heating the Fe-NCv-600 sample to 700°C by *in-situ* ETEM observation, which is followed by the evaporation and disappearance of Zn nanoparticles.

In **Fig. 2**, the Fe-NCv-700 sample is synthesized by pyrolysis of ZIF-8 and Fe(acac)₃@filter papers at 700°C **for 3 hours**. After heating at 700°C for 3 hours, the

unstable Zn nanoparticles evaporate and disappear. Therefore, no Zn-Zn bonds are observed in the Fe-NCv-700 sample after heating at 700°C for 3 hours.

Comment 5: In all the XPS spectra, no Fe was presented. The authors should use the method to identify it.

Reply: We sincerely appreciate the reviewer's suggestion to identify the XPS spectra of Fe element. The fitting results of the XPS spectra of Fe element in Fe-NCv and Fe-NC-900 samples are exhibited in **Figure R28-R29**. We refer the reference (*Nat. Mater.* **20** 1385-1391 (2021).) to analyze the Fe³⁺, Fe²⁺ components and the satellite peaks. We add the XPS spectra of Fe element in Fe-NCv and Fe-NC-900 samples in **Supplementary Figs. 14-19** and **32** of the revised **Supplementary Information**.

Figure R28. The fitting results of the XPS spectra of Fe element in Fe-NCv samples.

Figure R29. The fitting result of the XPS spectrum of Fe element in Fe-NC-900.

Comment 6: Similarly, Fe-NC-900 was found to be much active, while this material was either not well characterized so that it is confirmed that Fe-NC-900 did not have strong defect. The authors should check Raman spectra to get I_D/I_G .

Reply: We sincerely appreciate the reviewer's recommendation to check the Raman spectrum of Fe-NC-900 catalyst. As shown in **Figure R30**, the I_D/I_G value of Fe-NC-900 is 0.94, lower than that of Fe-NCv-900 catalyst with the I_D/I_G value of 1.07, indicating that the higher degree of disorder from carbon substrates of Fe-NCv-900 than that of Fe-NC-900 catalyst. We add the **Figure R30a** as **Supplementary Fig. 33** in the revised **Supplementary Information**.

Figure R30. The comparison of the Raman spectra of Fe-NC-900 and Fe-NCv-900 catalysts.

Comment 7: In DFT calculation, ZnO/graphene was used, which may not be correct for a comparison. The calculation should be ZnO@Fe-N₄.

Reply: We sincerely appreciate the reviewer's comment. In the DFT calculation, except for ZnO@graphene, we also simultaneously construct three ZnO@Fe-N₄/N-doped graphene hetero-structures, as shown in **Supplementary Figs. 60-62** (the **Supplementary Figs. 40-42** in the previous version). The formation of C-O bond between carbon-substrate and ZnO is the key step during formation of carbon-vacancy defects. As shown in **Fig. 6c**, compared with ZnO@graphene, ZnO@Fe-N₄/Cv-1, ZnO@Fe-N₄/Cv-2 and ZnO@Fe-N₄/Cv-3 have lower energy

changes for C-O bonding, indicating the easier formation of C-O bonding adjacent to Fe-N₄ sites on N-doped graphene than pure graphene and easier formation of carbon vacancies adjacent to Fe-N₄ sites.

Supplementary Fig. 60 (Supplementary Fig. 40 in the previous version). a-b, The top view and side view of the optimized structure of ZnO@Fe-N₄/N-doped graphene hetero-structure for synthesis of Fe-N₄/Cv-1 before and after C-O bonding.

Supplementary Fig. 61 (Supplementary Fig. 41 in the previous version). a-b, The top view and side view of the optimized structure of ZnO@Fe-N₄/N-doped graphene hetero-structure for synthesis of Fe-N₄/Cv-2 before and after C-O bonding.

Supplementary Fig. 62 (Supplementary Fig. 42 in the previous version). a-b, The top view and side view of the optimized structure of ZnO@Fe-N₄/N-doped graphene hetero-structure for synthesis of Fe-N₄/Cv-3 before and after C-O bonding.

Fig. 6c. The comparison of energy changes for C-O bonding of ZnO@graphene, ZnO@Fe-N₄/N-doped graphene hetero-structures.

Comment 8: It would be better that the authors discuss why high valent Fe was the active site and no other reaction route occurred.

Reply: We sincerely appreciate the reviewer's recommendation to discuss the high valent Fe as catalytic sites. Therefore, we perform the *in-situ* XANES and EXAFS measurements at Fe K-edge and revise the Manuscript as follows:

“To test the existence form of Fe catalytic sites from Fe-NCv-900 during

Fenton-like reaction, *in-situ* XANES and EXAFS measurements were performed. The Fe-NCv-900 catalyst on carbon paper was immersed into 10 μM phenol and 100 μM PMS aqueous solution during *in-situ* Fenton-like reaction. In **Fig. 5a**, the XANES spectra of Fe-NCv-900 and Fe-NCv-900 during *in-situ* Fenton-like reaction exhibited similar characteristic with that of FePc, with the pre-edge peak around 7114 eV assigned to the centrosymmetric Fe-N₄ square-planar structure (*Nat. Commun.* **11**, 4173 (2020).), indicating the Fe element existed as Fe ISAS during catalysis. As shown in the insert of **Fig. 5b**, the energy of the adsorption edge of Fe-NCv-900 during *in-situ* Fenton-like reaction exhibited an obvious positive shift compared to Fe-NCv-900. Besides, the white line intensity around 7135 eV of Fe-NCv-900 during *in-situ* Fenton-like reaction was higher than that of Fe-NCv-900, confirming the formation of high-valent iron-oxo species. (*Adv. Mater.* **34**, 2110653 (2022).) The corresponding Fourier-transform EXAFS (FT-EXAFS) spectra in R space were shown in **Fig. 5c**. During *in-situ* Fenton-like reaction, the intensity of the dominant peak of Fe-NCv-900 at around 1.5 Å obviously increased compared to Fe-NCv-900 catalyst, which was assigned to the *in-situ* formation of Fe-O bond from high-valent iron-oxo species. Compared with the WT contour plot of Fe-NCv-900 in **Fig. 5d**, the intensity of the dominant peak (4.5 Å⁻¹ in k space, 1.5 Å in R space) of Fe-NCv-900 during *in-situ* Fenton-like reaction in **Fig. 5e** obviously increased and the spatial distribution of the dominant peak during *in-situ* Fenton-like reaction moved towards higher k space, owing to the formation of Fe-O bonds from high-valent iron-oxo species during catalysis, which was similar to the Fe-O pathway from Fe₂O₃ in **Fig. 5f**. Compared with Fe₂O₃, the absence of Fe-O-Fe pathway (8.5 Å⁻¹, 3.0 Å) from Fe-NCv-900 during *in-situ* Fenton-like reaction demonstrated the sole existence of Fe-ISAS, revealing the excellent stability of Fe-ISAS from Fe-NCv-900 during catalysis.”

Fig. 5 The *in-situ* XANES and EXAFS measurements for Fenton-like reaction. **a** The XANES spectra of FePc, Fe-NCv-900, and Fe-NCv-900 during *in-situ* Fenton-like reaction. **b** The comparison of XANES spectra of Fe-NCv-900 and Fe-NCv-900 during *in-situ* Fenton-like reaction. **c** The corresponding FT-EXAFS spectra in R space of Fe-NCv-900 and Fe-NCv-900 during *in-situ* Fenton-like reaction. **d-f** The WT analysis of Fe-NCv-900, Fe-NCv-900 during *in-situ* Fenton-like reaction and Fe₂O₃.

“We further investigated the reactive species for Fenton-like reactions. In **Supplementary Fig. 49**, we explored the reactive oxygen species (ROS) in the Fe-NCv-900/PMS system by EPR experiments with 5,5-dimethyl-1-pyrroline N-oxide (DMPO) as a trapping agent. No signals appeared when PMS or Fe-NCv-900 solely existed while Fe-NCv-900/PMS system exhibited a strong characteristic signal of 5,5-dimethyl-1-pyrrolidone-2-oxyl (DMPOX) with peak intensity of 1:2:1:2:1:2:1 (*Adv. Mater.* **34**, 2110653 (2022)). Previous study suggested that DMPO could be oxidized into DMPOX by radical or non-radical pathway (*Adv. Mater.* **34**, 2110653 (2022); *Angew. Chem. Int. Ed.* **61**, e202200406 (2022); *Environ. Sci. Technol.* **54**, 3714-3724 (2020); *Appl. Catal. B: Environ.* **241**, 561-569 (2019)).

Supplementary Fig. 49. The EPR experiments with 5,5-dimethyl-1-pyrroline N-oxide (DMPO) as a trapping agent. **a**, Fe-NCv-900/PMS system ($[\text{catalyst}]_0 = 10$ mg/L, $[\text{DMPO}]_0 = 100$ mM, $[\text{PMS}]_0 = 100$ μM , initial pH = 7.0 ± 0.1). **b**, only Fe-NCv-900 ($[\text{catalyst}]_0 = 10$ mg/L, $[\text{DMPO}]_0 = 100$ mM, initial pH = 7.0 ± 0.1). **c**, only PMS ($[\text{DMPO}]_0 = 100$ mM, $[\text{PMS}]_0 = 100$ μM , initial pH = 7.0 ± 0.1).

Then we added 100 mM tert-butanol (TBA) or ethanol (EtOH) as quenching agents to scavenge radicals species (*Adv. Mater.* **34**, 2110653 (2022).) into Fe-NCv-900/PMS system. The degradation of phenol was not inhibited after adding TBA or EtOH, demonstrating radical was not ROS in this system (**Supplementary Fig. 50**).

Supplementary Fig. 50. The quenching experiments with TBA or EtOH as quenching agents to scavenge HO[•] and SO₄^{•-} radicals. ([catalyst]₀ = 10 mg/L, [Phenol]₀ = 10 μM, [PMS]₀ = 100 μM, initial pH = 7.0±0.1)

Except for radical as ROS, singlet oxygen (¹O₂) (*Angew. Chem. Int. Ed.* **61**, e202202338 (2022).) and high-valent metals species (*Adv. Mater.* **34**, 2110653 (2022); *Proc. Natl. Acad. Sci. USA.* **120**, e2219923120 (2023).) as ROS were reported for activation of PMS. When using 2,2,6,6-tetramethylpiperidine (TEMP) as spin-trapping agent for ¹O₂ (*Adv. Mater.* **34**, 2110653 (2022); *Angew. Chem. Int. Ed.* **61**, e202202338 (2022).), no signals were detected in **Supplementary Fig. 51**, excluding ¹O₂ as ROS.

Supplementary Fig. 51. The EPR spectrum with TEMP as spin trapping agents for singlet oxygen (¹O₂). ([catalyst]₀ = 10 mg/L, [TEMP]₀ = 100 mM, [PMS]₀ = 100 μM, initial pH = 7.0 ± 0.1)

In **Fig. 4g**, methyl phenyl sulfoxide (PMSO) as a chemical probe of high-valent iron-oxo species (*Adv. Mater.* **34**, 2110653 (2022).) was added in Fe-NCv-900/PMS system. The peak of methyl phenyl sulfone (PMSO₂) by high performance liquid chromatography (HPLC) measurement demonstrated the formation of high-valent iron-oxo species for activation of PMS. Therefore, we concluded that high-valent iron-oxo species was the ROS in Fe-NCv-900/PMS system, because PMSO was oxidized into PMSO₂ by high-valent iron-oxo species.”

Fig. 4g High performance liquid chromatography (HPLC) spectra with PMSO as a chemical probe of high-valent iron-oxo species.

Besides, additional inhibition experiment is performed to confirm the high-valent iron-oxo species as active sites. We add 100 mM Dimethyl sulfoxide (DMSO) as the inhibitor of high-valent iron-oxo species in Fe-NCv-900/PMS system for degradation of phenol. As reported by the reference (*Angew. Chem.* **117**, 7031-7034 (2005).), the DMSO can consume the high-valent iron-oxo species by oxygen-atom-transfer step.

As shown in **Figure R22**, after adding 100 mM Dimethyl sulfoxide (DMSO) as the inhibitor of high-valent iron-oxo species, the removal ratio of phenol is 42.0% after 5 min for degradation, with an obvious decline compared to the control group without DMSO (97% removal ratio of phenol after 5 min for degradation), indicating that the decreasing activity of Fe-NCv-900 is attributed to the inhibition of DMSO for high-valent iron-oxo species, which is consistent with previous studies (*Appl. Catal. B: Environ.* **305**, 123049 (2022); *Chem. Eng. J.* **427**, 130803 (2022).).

Figure R22. The inhibition experiment of the high-valent iron-oxo species by adding 100 mM Dimethyl sulfoxide (DMSO) as the inhibitor of high-valent iron-oxo species in Fe-NCv-900/PMS system for degradation of phenol.

Therefore, the high-valent iron-oxo species are demonstrated as the active sites during Fenton-like reaction for degradation of phenol by combination of *in-situ* XANES and EXAFS measurements, quenching experiments which exclude the radical and singlet oxygen ($^1\text{O}_2$) as ROS, and inhibition experiment by DMSO.

REVIEWERS' COMMENTS

Reviewer #1 (Remarks to the Author):

All my quires has been addressed in the revision, and it can be accepted.

Reviewer #2 (Remarks to the Author):

The revised manuscript could be accepted in the current form.